# Benzotriazine Di-Oxide Prodrugs for Exploiting Hypoxia and Low Extracellular pH in Tumors

**DOI:** 10.3390/molecules24142524

**Published:** 2019-07-10

**Authors:** Michael P. Hay, Hong Nam Shin, Way Wua Wong, Wan Wan Sahimi, Aaron T.D. Vaz, Pooja Yadav, Robert F. Anderson, Kevin O. Hicks, William R. Wilson

**Affiliations:** 1Auckland Cancer Society Research Centre, School of Medical Sciences, Faculty of Medical and Health Sciences, University of Auckland, Auckland 1142, New Zealand; 2Maurice Wilkins Centre for Molecular Biodiscovery, University of Auckland, Symonds St, Auckland 1142, New Zealand; 3School of Chemical Sciences, University of Auckland, Auckland 1142, New Zealand

**Keywords:** bioreductive prodrugs, benzotriazine di-oxides, tumor acidosis, tumor hypoxia, pH-dependent partitioning, radical chemistry, tirapazamine, SN30000, CEN-209, chlorambucil, WST-1

## Abstract

Extracellular acidification is an important feature of tumor microenvironments but has yet to be successfully exploited in cancer therapy. The reversal of the pH gradient across the plasma membrane in cells that regulate intracellular pH (pHi) has potential to drive the selective uptake of weak acids at low extracellular pH (pHe). Here, we investigate the dual targeting of low pHe and hypoxia, another key feature of tumor microenvironments. We prepared eight bioreductive prodrugs based on the benzotriazine di-oxide (BTO) nucleus by appending alkanoic or aminoalkanoic acid sidechains. The BTO acids showed modest selectivity for both low pHe (pH 6.5 versus 7.4, ratios 2 to 5-fold) and anoxia (ratios 2 to 8-fold) in SiHa and FaDu cell cultures. Related neutral BTOs were not selective for acidosis, but had greater cytotoxic potency and hypoxic selectivity than the BTO acids. Investigation of the uptake and metabolism of representative BTO acids confirmed enhanced uptake at low pHe, but lower intracellular concentrations than expected for passive diffusion. Further, the modulation of intracellular reductase activity and competition by the cell-excluded electron acceptor WST-1 suggests that the majority of metabolic reductions of BTO acids occur at the cell surface, compromising the engagement of the resulting free radicals with intracellular targets. Thus, the present study provides support for designing bioreductive prodrugs that exploit pH-dependent partitioning, suggesting, however, that that the approach should be applied to prodrugs with obligate intracellular activation.

## 1. Introduction

Hypoxia is a common feature of the tumor microenvironment [1] and contributes to tumor progression and resistance to therapy [2,3]. Low extracellular pH (pHe) results from the high glycolytic rates and slow clearance of metabolic acids in poorly perfused regions in tumors [4,5,6,7,8]. Extracellular acidification is an important contributor to tumor invasion and metastasis [9,10,11]. Like hypoxia, acidosis is a potentially exploitable difference between normal tissues and tumors that can be utilized in drug design. Tumor acidification is largely restricted to the extracellular compartment [12,13], and the resulting reversal in the trans-membrane pH gradient will drive the accumulation of weak acids by pH-dependent partitioning [12,14,15]; the predicted pKa dependence of this selective cellular uptake for passively transported weak acids is illustrated in Appendix A. The reversal of the pH gradient across the plasma membrane in regions of acidosis reflects active proton export by carbonic anhydrase IX and by monocarboxylate (lactate), bicarbonate, and proton transporters and their upregulation under hypoxia [16,17]. Much effort has been directed to inhibiting pH regulation to elicit tumor cell killing, but the redundancy of these transporters has led to the conclusion that “we still do not have an effective acid-mediated tumor cell death protocol” [17]. In addition, despite the surge in novel nanoparticle formulations designed to release active agents in tumors [18], pH selective nanoparticles have yet to make an impact clinically [19,20].

The selectivity of weak acids, e.g., chlorambucil (CHL, **1**), for cells at low pHe has been experimentally confirmed [21,22,23] but has not yet been exploited for targeting the tumor microenvironment. We sought to exploit this tumor selectivity for weak acids by redesigning a hypoxia-selective bioreductive prodrug, SN30000 (**2**), to accommodate a weak acid moiety. SN30000 is a second generation benzotriazine di-oxide that undergoes selective activation to a DNA-reactive radical species under hypoxia [24]. It was designed to be a more potent, selective analogue of the clinically evaluated tirapazamine (TPZ, **3**) [25,26,27], and to have improved extravascular transport properties [28]. SN30000 displays increased killing of hypoxic cells both in vitro and in vivo when compared to TPZ [29,30]. 

Our hypothesis is that using pH-driven targeting to deliver a potent cytotoxin, and superimposing this on hypoxic activation, will provide prodrugs with improved tumor selectivity. Here, we explore a series of benzotriazine di-oxide analogues featuring weakly acidic side chains and evaluate their cytotoxicity, cellular uptake, and metabolism under hypoxia and low extracellular pH.

## 2. Results

### 2.1. Prodrug Design and Synthesis

We designed a series of weak acid analogues (**4a**–**12a**) and the corresponding esters (**4b**–**12b**) based on the benzotriazine di-oxide (BTO) core of compounds **2** and **3** (Figure 1). The initial series (**4a**–**7a**) consisted of propionic acids with alkyl substituents on the benzo ring. These substituents were selected to modulate lipophilicity and one-electron reduction potentials, thereby affecting rates of diffusion and metabolic activation under hypoxia, thus modulating extravascular transport properties. We also prepared a corresponding series of 3-aminoalkanoic acids (**8a**–**12a**) for comparison.

A Stille reaction of the appropriate 3-iodobenzotriazine 1-oxides **13** gave the corresponding aldehydes **14** which were oxidized to the respective acids **15** (Scheme 1). Direct oxidation of the BTO 1-oxide acids **15** to the di-oxides **4a**–**7a** was not possible. However, protection of the acids **15** as the corresponding esters **16** allowed oxidation to the BTO 1,4-dioxides **4b**–**7b** and subsequent acidic hydrolyses gave BTO 1,4-dioxide alkanoic acids **4a**–**7a**.

Displacement of the 3-chloro BTO 1-oxides **17** with the appropriate aminoalkyl esters gave the esters **18** which were oxidized to the 1,4-dioxides **8b**–**12b** (Scheme 2). These esters were then hydrolyzed to the BTO 1,4-dioxide aminoalkanoic acids **8a**–**12a** in good yields, with the exception of **9a**, which was not isolated due to decomposition under the reaction conditions.

### 2.2. Prodrug Physicochemical Properties

The physicochemical properties of the compounds are summarized in Table 1. The BTO acids were consistently more soluble than the corresponding esters in culture medium at pH 7.4, reflecting the contribution of the alkanoic acid moiety which was predominantly ionized under these conditions (pH >> pKa). The pKa value for **7a** in 0.15 M KCl, measured by potentiometric titration, was 4.16 ± 0.01, which is similar to that calculated (pKa 4.14) in the software package Chemdraw v17.1.0 (PerkinElmer Informatics Inc., Cambridge, MA, USA). Therefore, we used Chemdraw to estimate pKa for the other acids, including chlorambucil, although we note that the latter value (pKa 4.62) is lower than the widely quoted [21,31,32] experimental estimate of 5.8, which, as originally reported, was potentially compromised by the precipitation of the free acid [33]. The calculated pKa values of the BTO acids covered a range from 3.7 to 4.5 and are thus in a range suitable for exploiting transmembrane pH differentials. These values are lower than the calculated pKa of 4.70 for long-chain (*n* > 10) alkanoic acids, indicating the modest effect of the BTO nucleus on pKa. However, the cLogP and cLogD values were substantially lower than for chlorambucil in all cases, with cLogD at pHe 7.4, less than −3 in four of the eight BTO acids, suggesting that passive diffusion through membranes could be slow. 

One-electron reduction potentials (*E*^0′^) were determined from the redox equilibria with methylviologen, MV^2+^, (*E*^0′^ = 447 ± 7 mV, [34]), dimethyldiquat, V21^2+^, (*E*^0′^ = 491 ± 6 mV, [35]), or triquat, TQ^2+^, (*E*^0′^ = −548 ± 7 mV [34]). The *E*^0′^ for **4a** was similar to that of SN30000, **2**, suggesting this BTO acid will be an efficient substrate for enzymatic one-electron reduction, even though its *E*^0′^ is 36 mV lower than the corresponding methyl ester (**4b**) reflecting the positive inductive effect of the sidechain and charge on the electronics of the benzotriazine nucleus. Substituting the 3-alkanoic sidechain with the 3-aminoalkanoic sidechain on the BTO 1,4-dioxide lowers the *E*^0′^, as does the addition of the cyclopentane ring. The combined effect is seen for **10a**, where the *E*^0′^ value is −565 mV, which is expected to be too low for significant bioreduction under hypoxia.

### 2.3. Free Radical Chemistry of BTO Acid ***7a***

The free radical chemistry of **7a** was investigated by pulse radiolysis and electron paramagnetic resonance (EPR). The pKa of the radical anion of **7a** (5.69 ± 0.15; Appendix A) was found to be similar to that of SN30000 (5.48 ± 0.26, [24]). The radical anion decayed slowly under anoxia at pH 7 with mixed kinetics (first-order rate constant of 78 ± 4 s^−1^, second-order rate constant of 3.8 ± 0.7 × 10^7^ M^−1^ s^−1^, Appendix A), broadly similar to SN30000 [24]. The second-order rate constant for oxidation of the radical anion of **7a** by O_2_, 5.39 ± 0.40 × 10^6^ M^−1^ s^−1^ (Appendix A), is consistent with the known dependence on *E*^0′^ for BTOs [37]. 

An EPR study using the spin trap *N*-*tert*-butyl-α-phenylnitrone (PBN) was undertaken to identify possible reactive radicals formed upon the anaerobic reduction of **7a** by a soluble form of NADPH-cytochrome P450 oxidoreductase (sPOR) (Figure 2). The spectrum showed a triplet of doublets with wide splitting. Simulation of this spectrum (WINSIM, National Institute of Environmental Health Sciences (NIEHS)) is represented by the combination of two species in the ratio 0.81:0.19 with hyperfine coupling constants (HFC) of (i) aH 3.6 G and aN 16.3 G, and (ii) aH 4.3 G and aN 16.0 G, respectively. The HFCs of the former, (i), indicate the presence of a general carbon-centered radical, while the latter, (ii), matches that of an aryl radical [38,39]. Hence, the formation of a highly reactive aryl radical is a likely candidate for a cytotoxin formed upon the one-electron reduction of **7a**, as was found for SN30000 [24]. A similar EPR spectrum was measured in the presence of 2 M DMSO (Appendix A), ruling out the hydroxyl radical being produced, which would be scavenged by the DMSO, leading to the methyl radical being spin-trapped by PBN and a changed EPR spectrum [40].

### 2.4. pH Regulation by SiHa and FaDu Cells under Acidosis

Prior to testing the pH dependence of the cytotoxicity of BTO acids, we evaluated the regulation of intracellular pH (pHi) when extracellular pH (pHe) was lowered from 7.4 to 6.5 in SiHa and FaDu cell cultures. We determined pHe with a pH microelectrode, and pHi by loading cells with non-fluorescent 2′,7′-bis-(2-carboxyethyl)-5-(and-6)-carboxyfluorescein acetoxymethyl ester (BCECF AM), which hydrolyses intracellularly to the cell-entrapped ratiometric pH-sensitive fluorescent probe BCECF. BCECF fluorescence emission at 535 nm is pH-independent for an excitation at 440 nm and pH-dependent for a 490 nm excitation [41,42,43]. Cells were loaded with 2 μM BCECF AM for 30 min in Hank’s balanced salt solution (HBSS) at pH 7.4 or 6.5, and fluorescence was measured with a microplate reader approximately 3 min after washing with a fresh medium at the same pH in order to remove any extracellular BCECF. pHi values, determined by calibration of the 440 nm/490 nm fluorescence ratio with digitonin-permeabilized cells, are shown in Table 2. Under oxic conditions (20% O_2_ gas phase), pHi was almost equal to pHe at the standard pHe of 7.4, for both cell lines, while when pHe was acutely lowered to 6.5, pHi was higher than pHe. The pHi values appeared to be somewhat lower under hypoxia (0.2% O_2_), although these were not significantly different from oxia (20% O_2_), with *P* values of 0.243 at pHe 7.4 and 0.192 at pHe 6.5. Similar shifts in pHi were observed under acidosis with cells under oxia, hypoxia (0.2% O_2_), or anoxia (<0.01% O_2_). Thus, when pHe was lowered by 0.9 units, pHi fell by approximately 0.4–0.5 units only. In a preliminary experiment, these ΔpHi values for UT-SCC-74B cells under 20% or 0.2% O_2_ were similar to those for SiHa cells in the same experiment (data not shown). Thus, the cell lines only partially controlled pHi under acute acidosis, which is consistent with other studies in low cell density cultures [44,45]. 

### 2.5. pH and Hypoxia Dependence of Prodrug Cytotoxicity

#### 2.5.1. pH Dependence of IC_50_ Values under Oxia and Anoxia

The inhibition of cell proliferation by the compounds was evaluated following 4 h of exposure under oxia (20% O_2_) or anoxia (<0.01% O_2_ gas phase) at pHe 7.4 and 6.5. A representative experiment with SiHa cells, illustrated in Figure 3, demonstrates that chlorambucil (**1**) is selective for low pHe, with pH cytotoxicity ratios (PCR) of approximately 7, but has no selectivity for anoxia versus oxia. Conversely, SN30000 (**2**) showed anoxia cytotoxicity ratios (ACR) of approximately 140, with little or no dependence on pHe. BTO acid **4a** demonstrates selectivity for both these features of the tumor microenvironment, although the PCR values (~2.5) were less than for chlorambucil and the ACR values (~8) were less than for SN30000. The potency of **4a** was low, even under the most favorable conditions (IC_50_ approximately 50-fold higher than for SN30000 at pHe 6.5 under anoxia). 

We evaluated the series of BTO acids and their corresponding esters in this assay, for both SiHa and FaDu cells, with chlorambucil (CHL, **1**) and SN30000 (**2**) used as reference compounds in each experiment. The results are summarized in Table 3 (SiHa) and Table 4 (FaDu), and for the corresponding esters in Table 5 (SiHa) and Table 6 (FaDu). CHL showed greater pH selectivity under oxic than anoxic conditions in SiHa cultures, but this difference was not observed in FaDu cultures. SN30000 (**2**) and tirapazamine (TPZ, **3**) showed consistently high selectivity for anoxia, with greater selectivity for SN30000 than TPZ, as reported for other cell lines [29,30,46,47]. Neither of these BTOs showed pH selectivity, except that SN30000 was significantly less potent at pHe 6.5 than 7.4 under anoxia in both cell lines. The BTO acids (**4a**–**12a**) showed PCR values >1 in almost all cases for which IC_50_ values could be determined within solubility constraints, with the exception of the relatively acidic (pKa 3.67) 3-aminoacetic acid **8a**, which lacked pH selectivity in both cell lines. As noted above for **4a**, the BTO acids all had low anoxia cytotoxicity ratio (ACR) values, with on-scale ratios ranging from 2 to 8.3 for SiHa and 3 to 5.6 for FaDu. The potencies of the BTO acids under anoxia at pHe 6.5 ranged from 97 to 3800 μM, broadly reflecting the *E^0′^* values with **4a** (*E*^0′^ −399 mV) being the most potent and **10a** (*E^0′^* −565 mV) one of the least potent, although **7a** (*E*^0′^ −463 mV, similar to TPZ) was even less potent than **10a**. Notably, these potencies were much lower than for TPZ and SN30000 (range of 2.9–8.6 μM), despite similar *E^0′^* values, suggesting that there may be an impediment to cytotoxicity of the acids that is not due to compromised metabolic one-electron reduction. Overall IC_50_ ratios, reflecting TME (tumor microenvironment) selectivity (TME ratio, TMR = oxic pHe 7.4/anoxic pHe 6.5), were consistently less for the BTO acids than the reference BTOs.

The corresponding esters (**4b**–**12b**) were distinctly different from the BTO acids, in that they lacked pH selectivity, with PCR values clustered around unity under both oxia and anoxia (Table 5 and Table 6). This suggests that the pHe selectivity of the BTO acids is likely due to pH-dependent partitioning, as expected. The anoxic potencies of the esters (IC_50_ values 1.4–20.4 in SiHa, 3.2–43.7 in FaDu) were markedly greater than for the BTO acids in each case, and their ACR values were generally higher, although the anoxic selectivity and overall TME selectivity was still inferior to TPZ and SN30000.

We also evaluated the growth inhibition of the head and neck squamous cell carcinoma line UT-SCC-74B by CHL (**1**), SN30000 (**2**) and BTO acid **4a** under oxia and anoxia at both pHe values (Appendix A). This gave similar results to the SiHa and FaDu cell lines, with CHL PCR values of 7.9 under oxia and 5.9 under anoxia, while pH selectivity was significant but lower than CHL for **4a** (PCR 2.3 under oxia and 2.9 under hypoxia). Again, **4a** showed much lower ACR than for SN30000 at either pHe.

#### 2.5.2. pH Dependence of IC_50_ Values under Chronic Hypoxia

There are relatively few viable cells in tumors at O_2_ concentrations as low as in the above anoxic experiments. In addition, extensive changes in gene expression occur under chronic exposure to intermediate O_2_ concentrations (chronic hypoxia), including the expression of proteins that regulate pHi [48]. Thus, we also determined the IC_50_ values of CHL, SN30000, and **4a** in SiHa cultures grown for 24 h under 0.2% O_2_ before and during exposure to the drugs for 4 h (Table 7). Under these chronic hypoxic conditions, cell attachment to the plastic was less effective than under oxia or acute anoxia, requiring minor modifications to the IC_50_ assay method (detailed in Section 4.10). Therefore, oxic and anoxic exposures were also tested in these same experiments. The 24 h adaptation to hypoxia had little effect on the subsequent growth of the cultures at either pHe value (Appendix A).

Despite the changes in methodology, the IC_50_ values for the oxic and anoxic exposures (Table 7) were generally in good agreement with the previous results (Table 3). Under 0.2% O_2_, the potency of SN30000 was lower than under anoxia at both pHe values, as expected, with HCR values of approximately 20, in contrast to ACR values in the range 46–156. However, the PCR values for CHL and **4a** were not markedly influenced by exposure to hypoxia (or anoxia), with no statistically significant differences in PCR between the three O_2_ concentrations (*P* > 0.5 by one-way ANOVA for all three compounds). This finding is consistent with the similar pHi values at each O_2_ concentration (Table 2), suggesting that pHi regulation in SiHa cells is not markedly improved under chronic hypoxia. 

### 2.6. pH Dependence of Cellular Uptake of SN30000 and BTO Acids

The above cytotoxicity studies were broadly consistent with the cellular uptake of the weakly acidic BTOs by pH-dependent partitioning, but the low potency of the BTO acids relative to neutral BTOs with similar *E^0′^* values suggests the low logD values of the acids might impede uptake. To investigate cellular uptake directly, we measured the intracellular and extracellular concentrations of SN30000 and three BTO acids (**4a, 7a, 11a**) in aerobic SiHa cell suspensions by HPLC (Figure 4). In these experiments, cell pellets were collected by centrifugation and ^3^H-mannitol was used as a cell-excluded marker to enable correction for the entrapped extracellular medium. The intracellular concentrations (Ci) of SN30000 were higher than the extracellular concentrations (Ce) at both 5 min and 60 min (Figure 4A), while the Ci of the BTO acids (Figure 4B–D) were much lower than the Ce. However, for the two BTO acids with measurable Ci/Ce ratios (**7a**,**11a**), uptake was enhanced at pHe 6.5 relative to pHe 7.4 (Figure 4C,D).

The observed Ci/Ce ratios and the expected values based on pH-dependent partitioning at equilibrium are shown in Table 8. The observed ratios were higher than predicted for SN30000 but lower for the BTO acids. The ratio of intracellular concentrations at the two pHe values were not accurately defined in most cases because of the large uncertainties arising from the subtraction of the drug in extracellular medium in the cell pellets when Ci was less than Ce. However, for two cases in which Ci could be reasonably estimated at both pHe values, the Ci was higher, at pH 6.5 by 3.9 ± 1.3-fold (**7a** at 5 min) and 4.4 ± 1.2-fold (**11a** at 60 min). In contrast, the Ci ratio at pHe 6.5/Ci at pHe 7.4 for SN30000 (**2**) was 1.09 ± 0.09 at 5 min and 0.95 ± 0.07 at 60 min. These ratios were broadly consistent with pH-dependent partitioning theory (Equations (2) and (3), Section 4.6) which gave Ci ratios (pHe 6.5/7.4) of 3.08 and 3.07 for **7a** and **11a,** respectively, and 0.51 for SN30000. This pH dependence of cell uptake is also broadly consistent with the pH dependence of cytotoxicity (PCR values 2–5) of **7a** and **11a** by SiHa cells (Table 3).

We next investigated the possible reasons for the lower-than-expected Ci/Ce values for the BTO acids. Given that the BCECF method reports an average pHi in the compartment(s) interrogated by the fluorogenic probe, we asked what pHi value would be needed to account for the low Ci/Ce ratios. For **7a**, the observed ratios (~0.1 at pHe 7.4 and ~0.7 at pHe 6.5) could be simulated when the pHi was set at 6.4 for both pHe values, which is implausibly low. Alternatively, the binding of these anionic compounds to bovine serum albumin in FBS could contribute [49] to this, and we have previously shown that the 1-oxide acid **15d**, which is a metabolite of SN30000 in mice, is >90% protein bound in mouse plasma [50]. Using the same equilibrium dialysis method, we found that **7a** at 100 μM is 62 ± 1% bound in FBS, but binding was not detectable in a culture medium with 5 or 10% FBS (Appendix A), and it thus unlikely to be responsible for low cell uptake. A third possibility is that efflux transporters could be responsible for the low Ci values, but in preliminary experiments we found no effect of overexpression from breast cancer resistance protein (BCRP) or P-glycoprotein (Pgp) efflux transporters (Appendix A) on sensitivity to **4a** or **7a** (or SN30000 (**2**)) in MDCK-11 cell lines under anoxia, and no effect from the pan-inhibition of monocarboxylate transporters (MCT) by phenylpyruvate [51] on sensitivity of anoxic SiHa cells to **4a** (Appendix A). We also evaluated the temperature dependence of uptake by SiHa cell suspensions. The Ci/Ce ratios for SN30000 (**2**) showed a minor increase at 4 °C, relative to 37 °C, but **4a** was again undetectable at either temperature (data not shown) which also argues against efflux transporters being responsible.

### 2.7. Anoxic Metabolism of SN30000 and BTO Acids: Metabolite Identification, and Quantitation

A further possible cause of low intracellular concentrations could be the metabolic consumption of the BTO acids, lowering the steady state Ci values. No metabolites of SN30000 or the BTO acids were observed in the above cellular uptake studies under 20% O_2_ (data not shown), making this unlikely. However, under anoxic conditions, the time-dependent formation of a less polar metabolite was observed in the extracellular medium in each case, as illustrated for **4a** in Figure 5A. These metabolites were not formed in any medium without cells under the same conditions and were identified as the corresponding *N*-1-oxides by their identical retention times and absorbance spectra to the authentic compounds (SN30000 1-oxide, **15a**, **15d**) 

The *N*-1-oxide metabolites of BTOs are known to have very low cytotoxic potencies [29,50,52,53], reflecting that the active cytotoxic species are the intermediate one-electron reduced radicals [24,52,54,55,56,57]. However, the 1-oxides are of interest in the present context as downstream markers of the reductive activation pathway. Quantitation of 1-oxide formation at 2 h (Figure 5B) demonstrated, surprisingly, that the metabolism by SiHa cells was independent of pHe, and that the rate was significantly higher for the BTO acids **4a** and **7a** than for SN30000, despite their low Ci values and low one-electron reduction potentials, as noted above. 

To investigate the one-electron reductive metabolism of the BTO acids further, we compared the anoxic metabolism between SiHa and SiHa/POR. The latter cell line over-expresses full-length NADPH-cytochrome P450 reductase (POR), which is the major one-electron reductase for SN30000 and other BTOs [30,58,59]. As expected, SiHa/POR metabolized SN30000 to its *N-*1-oxide more rapidly than the parental SiHa cells, but the rate of metabolism of **4a** and **7a** was essentially unchanged by the forced expression of POR (Figure 5C). A possible explanation for this is that the BTO acids are not substrates for POR in cells, despite the confirmed reduction of **7a** by sPOR in vitro (Figure 2), however, using anoxic S9 preparations from SiHa and SiHa/POR cells, we demonstrated that POR is able to reduce all three compounds to the corresponding 1-oxides when the plasma membrane barrier has been disrupted (Figure 5D). 

### 2.8. Cytotoxicity in Anoxic SiHa and SiHa/POR Cultures 

The lack of effect of POR over-expression on the reductive metabolism of the BTO acids in intact cells led us to compare the cytotoxicities of SN30000 (**2**), **4a**, and **7a** in SiHa/POR, and the parental line, under anoxia (Figure 6). For SN30000, the IC_50_ was 6.75-fold lower in the POR cell line, while for **4a** and **7a** the ratio was 3.4 and 2.7, respectively. This result, in conjunction with the lack of increase in total metabolism by the POR-expressing cells, suggested the hypothesis that the cytotoxicity of the BTO acids under anoxia is due to a minor component of the total reductive metabolism, with the majority of the metabolism to the 1-oxides not contributing to the cytotoxicity (these components are identified with intracellular reduction and extracellular reduction at the cell surface, respectively, in the model presented in Section 3). 

### 2.9. Cellular Uptake and Metabolism of BTO Ester **4b**

If the anionic sidechain of the BTO acids results in low intracellular concentrations, with reductive metabolism occurring largely at the cell surface, the corresponding esters would be expected to provide improved delivery to the intracellular compartment (consistent with their higher cytotoxic potencies, as noted in Section 2.5). To test this, we compared the metabolism of ester **4b** (Figure 7A) with its corresponding acid **4a** (Figure 7B) by monitoring extracellular concentrations in anoxic SiHa suspensions. The rates of metabolism of both compounds were unaffected by pHe, consistent with the results for **4a** in Figure 5B. The ester was metabolized more rapidly than the acid, with only a trace of acid **4a** detected transiently as an extracellular metabolite (Figure 7A). The major metabolites were the less polar 1-oxide, **15a**, and the 1-oxide of the ester, **16a**. In separate experiments with higher initial concentrations of **4b**, **15a** was again the main extracellular metabolite (Figure 7C, left). Notably, the intracellular concentrations of acid **4a** were extremely high, reaching a maximum of 4.1 mM at 30 min, while concentrations of its 1-oxide **15a** were approximately 10-fold lower (Figure 7C, right). Compound **4b** itself was not detected in the intracellular samples, presumably reflecting a lack of retention in cells and the very low intracellular volume, which will compromise analytical sensitivity (only 2 × 10^6^ cells were collected for each analysis in this experiment). These data demonstrate that the relatively polar acid **4a** is efficiently entrapped in cells if generated intracellularly from the ester, implying that the low intracellular concentration in cells incubated with **4a** itself is not due to efflux pumps.

### 2.10. WST-1 Inhibition of Anoxic Metabolism of BTO Acids

The above studies demonstrate that the BTO acids are efficiently reduced by anoxic cells, despite poor cellular uptake and lack of additional reduction when POR is over-expressed. One possibility for this is that the BTO acids are reduced to the 1-oxides by an extracellular one-electron reductase. To test this, we evaluated whether the cell-excluded disulfonic acid tetrazolium salt WST-1 could compete with BTO acids for one-electron reduction. WST-1 is a known electron acceptor for one-electron reduction by the plasma membrane electron transport system (PMET) [60] and is widely used as a cell viability assay, although the latter application uses 1-methoxyphenazine methylsulphate (mPMS) as an electron transfer intermediate. Given the possibility that mPMS could cross the plasma membrane itself, we used WST-1 without mPMS to ensure that the electron donor is a cell surface reductase. WST-1 potently inhibited the anoxic reduction of BTO acids **4a** and **7a** to their respective *N*-1-oxides, **15a** and **15d**, by SiHa cells, as assessed by HPLC, with IC_50_ values of 7.05 μM and 14.3 μM, respectively (Figure 8). In contrast, the metabolic reduction of SN30000 and TPZ was much less efficiently inhibited by WST-1, with IC_50_ values >1 mM (Figure 8). These results are consistent with most reductions of the BTO acids occurring extracellularly, while most reductions of SN30000 and TPZ occur intracellularly.

## 3. Discussion

In this study we illustrate the concept of designing prodrugs by superimposing selectivity for acidosis on selectivity for hypoxia in order to exploit these two important features of the tumor microenvironment (TME). Given that both features result from defective microvasculature in solid tumors, regions of hypoxia and low pHe at least partially overlap [61,62], indicating that such dual targeting has potential for enhancing the tumor selectivity of hypoxia activated prodrugs. We provide clear evidence that weakly acidic BTOs offer this dual selectivity in cell culture models, with consistently greater antiproliferative effects at pHe 6.5 than 7.4 (whether under oxia, hypoxia, or anoxia). Importantly, the BTO acids also demonstrate significant selectivity for low oxygen concentrations (Table 3, Table 4, Table 5, Table 6 and Table 7). This dual specificity for acidosis and hypoxia is illustrated by the summary of inferred mechanism of action of the BTO acids in Figure 9.

Neutral (BTO esters and TPZ) or weakly basic BTOs (SN30000) lack the modest pH selectivity of the BTO acids, but it is notable that the latter are much less potent under all conditions tested. In addition, the BTO acids show relatively modest enhancements in potency under anoxia or hypoxia, and their overall TME ratios (potency against hypoxic cells at pHe 6.5 versus oxic cells at pHe 7.4) are consistently less than neutral or basic compounds, at least for the BTO acids that are soluble enough to determine IC_50_ values under aerobic conditions. For these reasons we do not consider these compounds to be candidates for further development as anticancer agents, but investigation of the factor(s) that compromise their potency was warranted to better understand how this dual targeting approach could ultimately be applied more successfully.

The disappointingly low potency of the BTO acids can be ascribed, at least in part, to their low intracellular concentrations, which were less than extracellular concentrations, even at pHe 6.5. The intracellular concentrations were difficult to measure accurately because of the large correction required for the extracellular drug in the cell pellets, however, they were consistently lower than what was predicted by pH-dependent partitioning. However, the increase in Ci at pHe 6.5 relative to 7.4 was broadly in line with the theory (Table 8). Several mechanisms that could lower Ci below the equilibrium value were considered. The low pKa values of the acids (3.66–4.48) mean that at pHe 6.5, concentrations of the membrane-permeant (neutral) free acids would be ~0.1–1% of the total, while the anions are expected to have negligible membrane permeability [63], resulting in low logD values and slow kinetics of uptake into cells. Despite this, there was no obvious trend in Ci values between 5 min and 60 min (Table 8). Low steady state Ci could be maintained by an intracellular sink such as reductive metabolism, but under the oxic conditions used for the uptake studies we could not detect 1-oxides or any other metabolites. Reversible binding of the weak acids to albumin or other proteins in FBS could potentially limit free drugs available for trans-membrane diffusion, but we were not able to detect significant binding by equilibrium dialysis at the FBS concentrations used for the IC_50_ or cell uptake/metabolism studies. A limited investigation of efflux transporters, which could also maintain low intracellular concentrations, did not reveal a role for Pgp, BCRP, or MCTs (Appendix A). In addition, the intracellular generation of acid **4a** from ester **4b** resulted in very high Ci/Ce ratios of the acid (Figure 7), supporting a model in which the acids have very low membrane permeability. However, this does not preclude the involvement of unidentified transporters that become saturated at the millimolar intracellular concentrations of acid **4a** achieved from ester **4b**. We note that the roles of passive diffusion versus transporters for the movement of xenobiotics across the plasma membrane remains controversial [64,65], and that it is entirely possible that slow pH-dependent partitioning plus efflux by unidentified transporters both contribute to the low steady state Ci values of the BTO acids. These considerations suggest it may be productive to explore BTO acids with a moderately higher pKa (selectivity for acidosis would decrease with pKa values >6, Appendix A) and greater lipophilicity. 

While low intracellular concentrations restrict the potency of the BTO acids, this is clearly not the only limitation in the present series. The lack of the effect of forced expression of the one-electron reductase POR on the metabolism or cytotoxicity of the BTO acids under anoxia, relative to SN30000 (**2**) (Figure 5 and Figure 6), coupled with the inhibition of reduction of the 1-oxides by the extracellular electron acceptor WST-1 (Figure 8), suggests extensive reduction by extracellular reductase(s), as illustrated in Figure 9. The reductive metabolism of extracellular xenobiotics has little been studied, but the trans-plasma membrane electron transport (PMET) system is a candidate here. The components of this ubiquitous system vary between cell types but utilize intracellular NAD(P)H to reduce extracellular substrates such as WST-1 and ferricyanide via CoQ10-linked oxidoreductases [66,67]. The flavoreductase STEAP4 has been identified recently as capable of reducing the bioreductive prodrug tarloxotinib extracellularly [68]. In the case of BTO bioreductive drugs, the active cytotoxic metabolites are highly reactive DNA-damaging free radicals [54,56,57,69,70,71,72,73]. Both SN30000 [24] and BTO acid **7a** (Figure 2) form spin-trappable radicals with EPR features that are suggestive of an aryl radical and not the hydroxyl radical. Aryl radicals are powerful oxidants that induce DNA breaks [74,75,76], but their extreme instability would preclude DNA damage when formed extracellularly. Thus, the extracellular reduction of BTO acids is likely to be an unproductive metabolism, contributing to their low cytotoxic potencies, in contrast to the extracellular reduction of tarloxotinib, which generates a stable, membrane-permeant pan-ErbB inhibitor.

In conclusion, the present study has demonstrated the concept of the dual targeting of hypoxia and low extracellular pH in cell culture models and has identified the key features of the cellular pharmacology and mechanism of action of BTO acids in this context. This investigation suggests that to successfully exploit pH-dependent partitioning in this context, this will require application to bioreductive prodrugs that have good membrane permeability as free acids and that undergo one-electron reduction catalyzed only by intracellular enzymes. 

## 4. Materials and Methods

### 4.1. General Chemistry Procedures

All final products were analyzed by reverse-phase HPLC, (ZORBAX Eclipse XDB C8 5 μm column, 4.6 × 150 mm; Agilent Technologies, Santa Clara, CA, USA) using an Agilent Technologies 1260 Infinity chromatography system equipped with a diode-array absorbance detector. Mobile phases were gradients of 80% acetonitrile/20% H_2_O (*v*/*v*) in 45 mM of ammonium formate at a pH 3.5 at 0.8 mL/min. Final compound purity was determined by monitoring at 280–380 nm and was >95%, with the exception of **8b** (93.7%). Melting points were determined on an Electrothermal 2300 melting point apparatus (Cole-Palmer, Stone, UK). NMR spectra were obtained on a Bruker Avance 400 spectrometer (Bruker Biospin, Alexandria, Australia) at 400 MHz for ^1^H and 100 MHz for the ^13^C spectra. Spectra were obtained in CDCl_3_ unless noted otherwise. The chemical shifts and coupling constants were recorded in units of ppm and Hz, respectively. Low resolution mass spectra were gathered by the direct injection of methanolic solutions into an Agilent 6120 mass spectrometer using an atmospheric pressure chemical ionization (APCI) mode with a fragmentor voltage of 50 V and a drying gas temperature of 250 °C. High resolution mass spectra (HRMS) were measured on an Agilent Technologies 6530 Accurate-Mass Quadrupole Time of Flight (Q-TOF) LC/MS interfaced with an Agilent Jet Stream Electrospray Ionization (ESI) source, allowing the detection of positive or negative ions. Organic solutions were dried over Na_2_SO_4_ and solvents were evaporated under reduced pressure on a rotary evaporator. Thin-layer chromatography was carried out on aluminium-backed silica gel plates (Merck 60 F_254_, Merck KGaA, Darmstadt, Germany) with the visualization of components by UV light (254 nm) or exposure to I_2_. Column chromatography was carried out on silica gel (Merck 230–400 mesh). DCM refers to dichloromethane, DME refers to dimethoxyethane, DMF refers to dimethylformamide, DMSO refers to dimethyl sulfoxide, EtOAc refers to ethyl acetate, FBS refers to fetal bovine serum, MeOH refers to methanol, NADPH refers to nicotinamide adenine dinucleotide phosphate hydrogen, “pet. ether” refers to the petroleum ether boiling fraction of 40–60 °C, PBN refers to α-phenyl-*N*-*tert*-butylnitrone, THF refers to tetrahydrofuran, and TFAA refers to trifluoroacetic anhydride. 

Chlorambucil was obtained from Sigma-Aldrich (Merck KGaA, Darmstadt, Germany) while SN30000 (**2**) and its 1*N*-oxide [28] and TPZ (**3**) [77] were prepared as previously described.

### 4.2. Synthesis of BTO-3-Alkanoic Esters (***4b**–**7b***) and Acids (***4a**–**7a***)

#### 4.2.1. Stille Coupling of BTO Iodides (**13a**–**13d**) with Allyl Alcohol

*3-(3-Oxopropyl)benzo[e][1,2,4]triazine 1-oxide (***14a***).* Pd(OAc)_2_ (447 mg, 1.99 mmol) was added to a degassed mixture of 3-iodobenzo[*e*][1,2,4]triazine 1-oxide (**13a**) [78] (5.44 g, 19.9 mmol), NaHCO_3_ (3.68 g, 43.8 mmol), tetrabutylammonium chloride (5.53 g, 19.9 mmol), and allyl alcohol (6.80 mL, 99.6 mmol) in acetonitrile (200 mL), and the mixture was stirred at reflux temperature for 2 h. The mixture was cooled and the solvent was evaporated. The residue was purified by chromatography, eluting with a gradient (30–50%) of EtOAc/pet. ether, to give aldehyde **14a** (2.53 g, 62%) as a tan powder: mp (EtOAc/pet. ether) 86–88 °C; ^1^H-NMR δ 9.45 (s, 1 H, CHO), 8.45 (br d, *J* = 8.7 Hz, 1 H, H-8), 7.73–7.98 (m, 2 H, H-5, H-7), 7.70 (ddd, *J* = 8.5, 6.8 Hz, 1 H, H-6), 3.38 (dd, *J* = 6.9, 6.7 Hz, 2 H, CH_2_), 3.14 (dd, *J* = 6.9, 6.7 Hz, 2 H, CH_2_); MS *m/z* 204.1 (MH^+^, 100%). Anal. calcd for C_10_H_9_N_3_O_2_: C, 59.11, H, 4.46; N, 20.68. Found: C, 59.11; H, 4.42; N, 20.65%.

*7-Methyl-3-(3-oxopropyl)benzo[e][1,2,4]triazine 1-oxide (***14b***).* Similarly, the reaction of 3-iodo-7-methylbenzo[*e*][1,2,4]triazine 1-oxide (**13b**) [78] (1.85 g, 6.44 mmol) with Pd(Oac)_2_ (145 mg, 0.64 mmol), NaHCO_3_ (1.19 g, 14.2 mmol), tetrabutylammonium chloride (1.79 g, 6.44 mmol) and allyl alcohol (2.20 mL, 32.3 mmol) gave aldehyde **14b** (955 mg, 47%) as an grey solid: mp (EtOAc/pet. ether) 69–71 °C; ^1^H NMR δ 9.94 (s, 1 H, CHO), 8.23 (br s, 1 H, H-8), 7.85 (d, *J* = 8.6, 1.8 Hz, 1 H, H-5), 7.75 (dd, *J* = 8.6, 1.8 Hz, 1 H, H-6), 3.37 (t, *J* = 6.9 Hz, 2 H, CH_2_), 3.12 (t, *J* = 6.9 Hz, 2 H, CH_2_), 2.58 (s, 3 H, 7-CH_3_); MS *m/z* 218.1 (MH^+^, 100%). Anal. calcd for C_11_H_11_N_3_O_2_: C, 60.82, H, 5.10; N, 19.34. Found: C, 60.72; H, 5.01; N, 19.12%. 

*6-Methyl-3-(3-oxopropyl)benzo[e][1,2,4]triazine 1-Oxide (***14c***).* Similarly, reaction of 3-iodo-6-methylbenzo[*e*][1,2,4]triazine 1-oxide (**13c**) [78] (3.00 g, 10.5 mmol) with Pd(OAc)_2_ (234 mg, 1.05 mmol), NaHCO_3_ (1.93 g, 23.0 mmol), tetrabutylammonium chloride (2.89 g, 10.5 mmol) and allyl alcohol (3.56 mL, 52.3 mmol) gave aldehyde **14c** (1.40 g, 44%) as an cream powder: mp (EtOAc/pet. ether) 71–72 °C; ^1^H NMR δ 9.94 (s, 1 H, CHO), 8.32 (d, *J* = 8.8 Hz, 1 H, H-8), 7.72 (br s, 1 H, H-5), 7.50 (dd, *J* = 8.8, 1.7 Hz, 1 H, H-7), 3.36 (t, *J* = 6.9 Hz, 2 H, CH_2_), 3.12 (t, *J* = 7.1 Hz, 2 H, CH_2_), 2.59 (s, 3 H, 6-CH_3_); MS *m/z* 218.1 (MH^+^, 100%). Anal. calcd for C_11_H_11_N_3_O_2_: C, 60.82, H, 5.10; N, 19.34. Found: C, 60.94; H, 5.06; N, 19.59%.

*3-(1-Oxido-7,8-dihydro-6H-indeno[5,6-e][1,2,4]triazin-3-yl)propanal (***14d***).* Similarly, the reaction of 3-iodo-7,8-dihydro-6H-indeno[5,6-e][1,2,4]triazine 1-oxide **13d [78]** (7.15 g, 22.8 mmol) with Pd(OAc)_2_ (516 mg, 2.28 mmol), NaHCO_3_ (4.23 g, 50.0 mmol), tetrabutylammonium chloride (6.35 g, 22.8 mmol) and allyl alcohol (7.77 mL, 114 mmol) gave aldehyde **14d** (3.54 g, 64%) as an off white solid: mp (EtOAc/pet. ether) 72–74 °C; ^1^H NMR δ 9.93 (t, *J* = 0.9 Hz, 1 H, CHO), 8.25 (s, 1 H, H-9), 7.73 (s, 1 H, H-5), 3.35 (t, *J* = 7.0 Hz, 2 H, CH_2_), 3.07–3.14 (m, 6 H, H-6, H-8, CH_2_), 2.21 (p, *J* = 7.5 Hz, 2 H, H-7); ^13^C NMR δ 200.4, 163.9, 154.8, 149.1, 147.2, 132.3, 122.7, 114.2, 40.5, 33.1, 32.8, 29.4, 25.7; MS *m/z* 244.2 (MH^+^, 100%); HRMS (CI, CH_3_OH) calcd for C_13_H_14_N_3_O_2_ (MH^+^) *m/z* 244.1086, found 244.1088. Anal. calcd for C_13_H_13_N_3_O_2_: C, 64.2; H, 5.4; N, 17.3. Found: C, 63.9; H, 5.5; N, 17.0%.

#### 4.2.2. Oxidation of Aldehydes (**14a**–**14d**).

*3-(2-Carboxyethyl)benzo[e][1,2,4]triazine 1-Oxide (***15a***).* A solution of NaClO_2_ (1.59 g, 14.1 mmol) in water (20 mL) was added dropwise to a stirred mixture of aldehyde **14a** (1.91 g, 9.39 mmol) in MeCN (80 mL) and KH_2_PO_4_ (319 mg, 2.35 mmol) in water (10 mL) and 35% H_2_O_2_ (1.15 mL, 11.3 mmol) at 5 °C and the mixture was stirred at 5 °C for 3 h. The mixture was acidified with aqueous HCl (1 M) and extracted with EtOAc (3 × 50 mL). The combined organic fraction was extracted with an aqueous NaOH solution (0.1 M, 3 × 50 mL). The aqueous fraction was acidified with an aqueous HCl solution (1 M) and extracted with EtOAc (3 × 50 mL). The organic fraction was dried and the solvent was evaporated. The residue was purified by chromatography, eluting with a gradient (50–100%) of EtOAc/pet. ether, to give acid **15a** (1.82 g, 88%) as a cream powder: mp 152–154 °C (dec.); ^1^H NMR [(CD_3_)_2_SO] δ 12.52 (br s, 1 H, CO_2_H), 8.37 (dd, *J* = 8.7, 0.8 Hz, 1 H, H-8), 8.08 (ddd, *J* = 8.7, 6.8, 1.4 Hz, 1 H, H-7), 8.03 (dd, *J* = 8.4, 1.4 Hz, 1 H, H-5), 7.89 (ddd, *J* = 8.4, 6.8, 1.4 Hz, 1 H, H-6), 3.19 (t, *J* = 7.1 Hz, 2 H, CH_2_), 2.83 (t, *J* = 7.1 Hz, 2 H, CH_2_); MS *m/z* 204.1 (MH^+^, 100%). Anal. calcd for C_10_H_9_N_3_O_3_: C, 54.79; H, 4.14; N, 19.17. Found: C, 55.06; H, 4.00; N, 19.31%.

*3-(2-Carboxyethyl)-7-methylbenzo[e][1,2,4]triazine 1-Oxide (***15b***).* Similarly, the reaction of NaClO_2_ (0.70 g, 6.19 mmol) with aldehyde **14b** (0.90 g, 4.12 mmol), KH_2_PO_4_ (140 mg, 1.03 mmol) and 30% H_2_O_2_ (0.51 mL, 4.94 mmol) gave acid **15b** (771 mg, 80%) as white needles: mp 191–193 °C (dec.); ^1^H NMR [(CD_3_)_2_SO] δ 12.24 (br s, 1 H, CO_2_H), 8.18 (br s, 1 H, H-8), 7.90–7.94 (m, 2 H, H-5, H-6), 3.17 (br t, *J* = 7.1 Hz, 2 H, CH_2_), 2.82 (br t, *J* = 7.1 Hz, 2 H, CH_2_), 2.55 (s, 3 H, 7-CH_3_); MS *m/z* 234.1 (MH^+^, 100%). Anal. calcd for C_11_H_11_N_3_O_3_: C, 56.65; H, 4.75; N, 18.02. Found: C, 56.73; H, 4.80; N, 18.16%.

*3-(2-Carboxyethyl)-6-methylbenzo[e][1,2,4]triazine 1-Oxide (***15c***).* Similarly, the reaction of NaClO_2_ (1.00 g, 8.84 mmol) with aldehyde **14c** (1.28 g, 5.89 mmol), KH_2_PO_4_ (200 mg, 1.47 mmol) and 30% H_2_O_2_ (0.72 mL, 7.07 mmol) gave acid **15c** (1.32 g, 82%) as a white powder: mp 170–171 °C; ^1^H NMR [(CD_3_)_2_SO] δ 12.24 (br s, 1 H, CO_2_H), 8.32 (d, *J* = 8.8 Hz, 1 H, H-8), 7.74 (br s, 1 H, H-5), 7.51 (dd, *J* = 8.8, 1.7 Hz, 1 H, H-7), 3.35 (t, *J* = 7.1 Hz, 2 H, CH_2_), 3.02 (t, *J* = 7.1 Hz, 2 H, CH_2_), 2.59 (s, 3 H, 6-CH_3_); MS *m/z* 234.1 (MH^+^, 100%). Anal. calcd for C_11_H_11_N_3_O_3_: C, 56.65; H, 4.75; N, 18.02. Found: C, 56.92; H, 4.69; N, 18.25%.

*3-(2-Carboxyethyl)-7,8-dihydro-6H-indeno[5,6-e][1,2,4]triazine 1-Oxide (***15d***).* Similarly, the reaction of NaClO_2_ (222 mg, 1.96 mmol) with 3-(1-oxido-7,8-dihydro-6*H*-indeno[5,6-*e*][1,2,4] triazin-3-yl)-1-propanal **14d** (341 mg, 1.40 mmol) gave acid **15d** (261 mg, 72%) as a white powder: mp 215 °C (dec.); ^1^H NMR δ 8.24 (s, 1 H, H-9), 7.74 (s, 1 H, H-5), 3.34 (br t, *J* = 7.0 Hz, 2 H, CH_2_), 3.08–3.14 (m, 4 H, H-6, H-8), 3.01 (t, *J* = 7.0 Hz, 2 H, CH_2_), 2.22 (p, *J* = 7.4 Hz, 2 H, H-7), OH not observed; ^13^C NMR δ 176.4, 164.0, 155.2, 149.4, 147.2, 132.6, 122.8, 114.6, 33.3, 33.0, 31.4, 31.0, 25.9; MS *m/z* 260.3 (MH^+^, 100%); Anal. calcd for C_13_H_13_N_3_O_3_: C, 60.22; H, 5.05; N, 16.21. Found: C, 60.46; H, 5.08; N, 16.34%.

#### 4.2.3. Esterification of Acids (**15a**–**15d**).

*3-(3-Methoxy-3-oxopropyl)benzo[e][1,2,4]triazine 1-Oxide (***16a***).* cH_2_SO_4_ (cat., 0.3 mL) was added to a stirred suspension of acid **15a** (2.28 g, 10.4 mmol) in dry MeOH (100 mL) and the mixture was stirred at reflux temperature for 6 h. The mixture was cooled to 20 °C and the solvent was evaporated. The residue was partitioned between EtOAc (100 mL) and water (100 mL). The organic fraction was washed with an aqueous NaOH solution (0.1 M, 50 mL), then washed with water (25 mL), then washed with brine (30 mL), then dried, after which the solvent was evaporated. The residue was purified by chromatography, eluting with 50% EtOAc/pet. ether, to give ester **16a** (2.267 g, 94%) as a white powder: mp (EtOAc/pet. ether) 45–47 °C; ^1^H NMR δ 8.44 (dd, *J* = 8.6, 1.2 Hz, 1 H, H-8), 7.97 (dd, *J* = 8.7, 1.4 Hz, 1 H, H-5), 7.93 (ddd, *J* = 8.6, 6.9, 1.3 Hz, 1 H, H-7), 7.70 (ddd, *J* = 8.4, 6.9, 1.4 Hz, 1 H, H-6), 3.71 (s, 3 H, OCH_3_), 3.37 (t, *J* = 7.2 Hz, 2 H, CH_2_), 2.99 (t, *J* = 7.2 Hz, 2 H, CH_2_); MS *m/z* 234.2 (MH^+^, 100%). Anal. calcd for C_11_H_11_N_3_O_3_: C, 56.65; H, 4.75; N, 18.02. Found: C, 56.86; H, 4.56; N, 18.21%.

*3-(3-Methoxy-3-oxopropyl)-7-methylbenzo[e][1,2,4]triazine 1-Oxide (***16b***).* Similarly, the reaction of cH_2_SO_4_ (cat., 0.2 mL) with acid **15b** (730 mg, 3.13 mmol) in dry MeOH (50 mL) gave ester **16b** (740 mg, 96%) as a white powder: mp (EtOAc/pet. ether) 130–132 °C; ^1^H NMR δ 8.24 (br s, 1 H, H-8), 7.86 (d, *J* = 8.6 Hz, 1 H, H-5), 7.74 (dd, *J* = 8.6, 1.8 Hz, 1 H, H-6), 3.70 (s, 3 H, OCH_3_), 3.35 (t, *J* = 7.2 Hz, 2 H, CH_2_), 2.98 (t, *J* = 7.2 Hz, 2 H, CH_2_), 2.58 (s, 3 H, 7-CH_3_); MS *m/z* 248.2 (MH^+^, 100%). Anal. calcd for C_12_H_13_N_3_O_3_: C, 58.29; H, 5.30; N, 16.99. Found: C, 58.18; H, 5.31; N, 17.10%.

*3-(3-Methoxy-3-oxopropyl)-6-methylbenzo[e][1,2,4]triazine 1-Oxide (***16c***).* Similarly, the reaction of cH_2_SO_4_ (cat., 0.3 mL) with acid **15c** (1.09 g, 4.67 mmol) in dry MeOH (50 mL) gave ester **16c** (1.12 g, 97%) as white needles: mp (EtOAc/pet. ether) 91–93 °C; ^1^H NMR δ 8.32 (d, *J* = 8.8 Hz, 1 H, H-8), 7.73 (br s, 1 H, H-5), 7.50 (dd, *J* = 8.8, 1.7 Hz, 1 H, H-7), 3.70 (s, 3 H, OCH_3_), 3.35 (t, *J* = 7.2 Hz, 2 H, CH_2_), 2.97 (t, *J* = 7.2 Hz, 2 H, CH_2_), 2.59 (s, 3 H, 6-CH_3_); MS *m/z* 248.2 (MH^+^, 100%). Anal. calcd for C_12_H_13_N_3_O_3_: C, 58.29; H, 5.30; N, 16.99. Found: C, 58.18; H, 5.38; N, 17.18%.

*3-(3-Methoxy-3-oxopropyl)-7,8-dihydro-6H-indeno[5,6-e][1,2,4]triazine 1-Oxide (***16d***).* Similarly, the reaction of cH_2_SO_4_ (cat., 3 drops) with acid **15d** (330 mg, 1.27 mmol) in dry MeOH (30 mL) gave ester **16d** (277 mg, 80%) as a white powder: mp (EtOAc/pet. ether) 119–120 °C; ^1^H NMR δ 8.25 (s, 1 H, H-9), 7.74 (s, 1 H, H-5), 3.70 (s, 3 H, OCH_3_), 3.34 (t, *J* = 7.3 Hz, 2 H, CH_2_), 3.08–3.14 (m, 4 H, H-6, H-8), 2.96 (t, *J* = 7.3 Hz, 2 H, CH_2_CO), 2.21 (p, *J* = 7.5 Hz, 2 H, H-7); MS *m/z* 274.3 (MH^+^, 100%). Anal. calcd for C_14_H_15_N_3_O_3_: C, 61.53; H, 5.53; N, 15.38. Found: C, 61.64; H, 5.48; N, 15.44%.

#### 4.2.4. Oxidation of Benzotriazine 1-Oxides (**16a**–**16d**)

*3-(3-Methoxy-3-oxopropyl)benzo[e][1,2,4]triazine 1,4-Dioxide (***4b***).* H_2_O_2_ (30%, 9.8 mL, 95.1 mmol) was added dropwise to a stirred solution of TFAA (14.5 mL, 10.5 mmol) in DCM (30 mL) at 0–5 °C and the mixture was stirred at 20 °C for 1 h. This mixture was added dropwise to a stirred solution of ester **16a** (2.218 g, 9.51 mmol) in DCM (30 mL) at 0 °C and the mixture was stirred at 20 °C for 24 h. The mixture was cooled to 0 °C and DMSO (6.75 mL, 95.1 mmol) was added and the mixture was stirred for 5 min. The mixture was carefully neutralized with a concentrated NH_3_ solution and the mixture was washed with water (3 × 30 mL), then dried, after which the solvent evaporated was. The residue was purified by chromatography, eluting with a gradient (50–100%) of EtOAc/pet. ether, to give di-oxide **4b** (1.60 g, 68%) as a yellow powder: mp (EtOAc/pet. ether) 89–91 °C; ^1^H NMR δ 8.53 (dd, *J* = 8.7, 0.8 Hz, 1 H, H-8), 8.47 (dd, *J* = 8.7, 0.8 Hz, 1 H, H-5), 8.02 (ddd, *J* = 8.7, 7.1, 1.3 Hz, 1 H, H-7), 7.86 (ddd, *J* = 8.7, 7.1, 1.3 Hz, 1 H, H-6), 3.71 (s, 3 H, OCH_3_), 3.49 (t, *J* = 7.1 Hz, 2 H, CH_2_), 2.98 (t, *J* = 7.1 Hz, 2 H, CH_2_); ^13^C NMR [(CD_3_)_2_SO] δ 172.5, 154.1, 139.8, 135.7, 134.9, 132.1, 121.8, 119.7, 52.2, 28.8, 25.7; MS *m/z* 250.2 (MH^+^, 100%). Anal. calcd for C_11_H_11_N_3_O_4_: C, 53.01; H, 4.45; N, 16.86. Found: C, 53.17; H, 4.38; N, 16.93%.

*3-(3-Methoxy-3-oxopropyl)-7-methylbenzo[e][1,2,4]triazine 1,4-Dioxide (***5b***).* Similarly, the reaction of H_2_O_2_ (30%, 3.0 mL, 28.6 mmol) and TFAA (4.37 mL, 31.4 mmol) with ester **16b** (706 mg, 2.86 mmol) gave starting material **16b** (44 mg, 6%), spectroscopically identical to above, and di-oxide **5b** (586 mg, 78%) as a yellow powder: mp (EtOAc/pet. ether) 165–167 °C; ^1^H NMR δ 8.40 (d, *J* = 8.8 Hz, 1 H, H-5), 8.25 (br s, 1 H, H-8), 7.82 (dd, *J* = 8.8, 1.5 Hz, 1 H, H-6), 3.71 (s, 3 H, OCH_3_), 3.48 (t, *J* = 7.1 Hz, 2 H, CH_2_), 2.97 (t, *J* = 7.1 Hz, 2 H, CH_2_), 2.63 (s, 3 H, 7-CH_3_); ^13^C NMR δ 172.5, 153.4, 143.8, 138.2, 137.7, 134.7, 120.5, 119.5, 52.2, 28.8, 25.6, 22.0; MS *m/z* 264.1 (MH^+^, 100%). Anal. calcd for C_12_H_13_N_3_O_4_: C, 54.75; H, 4.98; N, 15.96. Found: C, 54.83; H, 4.93; N, 16.23%.

*3-(3-Methoxy-3-oxopropyl)-6-methylbenzo[e][1,2,4]triazine 1,4-Dioxide (***6b***).* Similarly, the reaction of H_2_O_2_ (30%, 4.0 mL, 38.4 mmol) and TFAA (5.88 mL, 42.3 mmol) with ester **16c** (950 mg, 3.84 mmol) gave starting material **16c** (81 mg, 9%), spectroscopically identical to above, and di-oxide **6b** (812 mg, 80%) as a yellow powder: mp (EtOAc/pet. ether) 166–168 °C; ^1^H NMR [(CD_3_)_2_SO] δ 8.26 (d, *J* = 8.9 Hz, 1 H, H-8), 8.18 (br s, 1 H, H-5), 7.78 (dd, *J* = 8.9, 1.6 Hz, 1 H, H-7), 3.63 (s, 3 H, OCH_3_), 3.27 (dt, *J* = 7.4, 7.1 Hz, 2 H, CH_2_), 2.97 (dt, *J* = 7.4, 7.1 Hz, 2 H, CH_2_), 2.61 (s, 3 H, 6-CH_3_); ^13^C NMR [(CD_3_)_2_SO] δ 172.2, 153.3, 147.4, 139.1, 133.9, 132.8, 120.8, 117.5, 51.6, 28.2, 25.2, 21.6; MS *m/z* 264.1 (MH^+^, 100%). Anal. calcd for C_12_H_13_N_3_O_4_: C, 54.75; H, 4.98; N, 15.96. Found: C, 54.51; H, 5.05; N, 16.22%.

*3-(3-Methoxy-3-oxopropyl)-7,8-dihydro-6H-indeno[5,6-e][1,2,4]triazine 1,4-Dioxide (***7b***).* Similarly, the reaction of H_2_O_2_ (30%, 1.25 mL, 10.0 mmol) and TFAA (1.26 mL, 9.1 mmol) with ester **16d** (249 mg, 0.91 mmol) gave di-oxide **7b** (200 mg, 76%) as a yellow powder: mp (EtOAc/pet. ether) 170–171 °C; ^1^H NMR δ 8.31 (s, 1 H, H-9), 8.25 (s, 1 H, H-5), 3.71 (s, 3 H, OCH_3_), 3.48 (t, *J* = 7.3 Hz, 2 H, CH_2_), 3.12–3.20 (m, 4 H, H-6, H-8), 2.97 (t, *J* = 7.2 Hz, 2 H, CH_2_CO), 2.26 (p, *J* = 7.5 Hz, 2 H, H-7); ^13^C NMR δ 172.6, 155.4, 153.2, 151.0, 139.2, 134.1, 116.1, 114.0, 52.2, 33.6, 33.1, 28.9, 25.8, 25.7; MS *m/z* 290.3 (MH^+^, 100%). Anal. calcd for C_14_H_15_N_3_O_4_: C, 58.13; H, 5.23; N, 14.53. Found: C, 57.91; H, 5.20; N, 14.44%.

#### 4.2.5. Hydrolysis of Esters (**4b**–**7b**)

*3-(2-Carboxyethyl)benzo[e][1,2,4]triazine 1,4-Dioxide (***4a***).* A solution of HCl in dioxane (4 M, 52 mL, 209 mmol) was added dropwise to a stirred suspension of ester **4b** (1.30 g, 5.22 mmol) in dioxane (30 mL) and water (1 mL) and the mixture was stirred at 20 °C for 5 d. The mixture was diluted with water (200 mL), extracted with CHCl_3_ (3 × 100 mL), then the combined organic fraction was dried and the solvent was evaporated. The residue was purified by chromatography, eluting with a gradient (0–5%) of MeOH/DCM, to give starting ester **4b** (390 mg, 30%), spectroscopically identical to above, and acid **4a** (728 mg, 59%) as a yellow powder: mp (EtOAc) 157 °C (dec.); ^1^H NMR [(CD_3_)_2_SO] δ 12.35 (br s, 1 H, CO_2_H), 8.35–8.40 (m, 2 H, H-5, H-8), 8.12 (br ddd, *J* = 8.6, 7.1, 1.4 Hz, 1 H, H-7), 7.96 (br ddd, *J* = 8.6, 7.1, 1.4 Hz, 1 H, H-6), 3.23 (br t, *J* = 7.2 Hz, 2 H, CH_2_), 2.77 (br t, *J* = 7.2 Hz, 2 H, CH_2_); ^13^C NMR [(CD_3_)_2_SO] δ 173.2, 153.5, 139.3, 135.7, 134.4, 132.1, 121.1, 118.8, 28.4, 25.3; MS *m/z* 276.3 (MH^+^, 100%). Anal. calcd for C_10_H_9_N_3_O_4_: C, 51.07; H, 3.86; N, 17.87. Found: C, 51.10; H, 3.80; N, 17.58%. HPLC purity 95.1%.

*3-(2-Carboxyethyl)-7-methylbenzo[e][1,2,4]triazine 1,4-Dioxide (***5a***).* Similarly, the reaction of HCl in dioxane (4 M, 25 mL, 100 mmol) with ester **5b** (522 mg, 1.98 mmol) gave starting ester **5b** (249 mg, 48%), spectroscopically identical to above, and acid **5a** (200 mg, 40%) as a yellow powder: mp (EtOAc/MeOH) 181–183 °C (dec.); ^1^H NMR [(CD_3_)_2_SO] δ 12.34 (br s, 1 H, CO_2_H), 8.26 (d, *J* = 8.8 Hz, 1 H, H-5), 8.18 (br s, 1 H, H-8), 7.95 (dd, *J* = 8.8, 1.6 Hz, 1 H, H-6), 3.22 (t, *J* = 7.1 Hz, 2 H, CH_2_), 2.77 (t, *J* = 7.2 Hz, 2 H, CH_2_), 2.57 (s, 3 H, 7-CH_3_); ^13^C NMR [(CD_3_)_2_SO] δ 173.2, 152.9, 143.1, 137.7, 137.5, 134.2, 119.6, 118.6, 28.4, 25.2, 21.1; MS *m/z* 250.1 (MH^+^, 100%). Anal. calcd for C_11_H_11_N_3_O_4_·¼CH_3_OH: C, 52.07; H, 4.94; N, 15.84. Found: C, 52.10; H, 4.80; N, 16.19%. HPLC purity 99.1%.

*3-(2-Carboxyethyl)-6-methylbenzo[e][1,2,4]triazine 1,4-Dioxide (***6a***).* Similarly, the reaction of HCl in dioxane (4 M, 34 mL, 136 mmol) with ester **6b** (714 mg, 2.71 mmol) gave starting ester **6b** (289 mg, 40%), spectroscopically identical to above, and acid **6a** (385 mg, 57%) as a yellow powder: mp (EtOAc) 173–175 °C (dec.); ^1^H NMR [(CD_3_)_2_SO] δ 12.35 (br s, 1 H, CO_2_H), 8.25 (d, *J* = 8.8 Hz, 1 H, H-8), 8.18 (br s, 1 H, H-5), 7.79 (dd, *J* = 8.8, 1.6 Hz, 1 H, H-7), 3.23 (t, *J* = 7.2 Hz, 2 H, CH_2_), 2.76 (t, *J* = 7.2 Hz, 2 H, CH_2_), 2.60 (s, 3 H, 6-CH_3_); ^13^C NMR [(CD_3_)_2_SO] δ 173.2, 153.6, 147.4, 139.1, 133.8, 132.8, 120.8, 117.5, 28.4, 25.3, 21.6; MS *m/z* 250.1 (MH^+^, 100%). Anal. calcd for C_11_H_11_N_3_O_4_: C, 53.01; H, 4.45; N, 16.86. Found: C, 52.94; H, 4.47; N, 16.80%. HPLC purity 99.6%.

*3-(2-Carboxyethyl)-7,8-dihydro-6H-indeno[5,6-e][1,2,4]triazine 1,4-Dioxide (***7a***).* Similarly, the reaction of HCl in dioxane (4 M, 4.32 mL, 17.3 mmol) with ester **7b** (100 mg, 0.34 mmol) gave starting ester **7b** (13 mg, 13%), spectroscopically identical to above, and acid **7a** (60 mg, 64%) as a yellow powder: mp (EtOAc) 159 °C (dec.); ^1^H NMR δ 12.34 (br s, 1 H, CO_2_H), 8.18 (s, 1 H, H-9), 8.17 (s, 1 H, H-5), 3.22 (t, *J* = 7.2 Hz, 2 H, CH_2_), 3.07–3.16 (m, 4 H, H-6, H-8), 2.76 (t, *J* = 7.2 Hz, 2 H, CH_2_CO), 2.13 (p, *J* = 7.5 Hz, 2 H, H-7); ^13^C NMR δ 173.2, 154.6, 152.7, 150.2, 138.7, 133.5, 115.2, 113.1, 32.8, 32.3, 28.5, 25.2 (2); MS *m/z* 276.3 (MH^+^, 100%). Anal. calcd for C_13_H_13_N_3_O_4_: C, 56.72; H, 4.76; N, 15.27. Found: C, 57.00; H, 4.73; N, 15.08%. HPLC purity 99.7%.

#### 4.2.6. Synthesis of BTO-3-aminoalkanoic Esters (**8b**–**12b**) and Acids (**8a**–**12a**)

*Ethyl [(1-Oxido-1,2,4-benzotriazin-3-yl)amino]acetate (***18a***).* A mixture of 3-chloro-1,2,4-benzotriazine 1-oxide (**17a**)[78] ( 2.02 g, 11.1 mmol), glycine ethyl ester hydrochloride (2.33 g, 16.7 mmol), and Et_3_N (4.2 mL, 30 mmol) in DME (100 mL) was heated at reflux temperature for 6 h. The solvent was evaporated and the residue was partitioned between DCM/water (200 mL), then the aqueous fraction extracted with DCM (2 × 50 mL), then the combined organic fraction was dried and the solvent was evaporated. The residue was purified by chromatography, eluting with 10% EtOAc/DCM, to give the ester **18a** (2.75 g, 99%) as a yellow solid: mp (EtOAc/DCM) 136–138 °C; ^1^H NMR δ 8.27 (dd, *J* = 8.6, 1.0 Hz, 1 H, H 8), 7.72 (ddd, *J* = 8.5, 7.0, 1.4 Hz, 1 H, H 6), 7.62 (dd, *J* = 8.5, 1.0 Hz, 1 H, H 5), 7.34 (ddd, *J* = 8.6, 7.0, 1.0 Hz, 1 H, H 7), 5.87 (br s, 1 H, NH), 4.30 (d, *J* = 5.7 Hz, 2 H, CH_2_N), 4.26 (q, *J* = 7.2 Hz, 2 H, CH_2_O), 1.31 (t, *J* = 7.2 Hz, 3 H, CH_3_); ^13^C NMR δ 169.9, 158.4, 148.5, 135.6, 131.2, 126.7, 125.5, 120.4, 61.6, 43.2, 14.2; MS *m/z* 249.2 (MH^+^, 100%). Anal. calcd for C_11_H_12_N_4_O_3_: C, 53.2; H, 4.9; N, 22.6. Found: C, 53.4; H, 5.0; N, 22.6%.

*3-((2-methoxy-2-oxoethyl)amino)-7,8-dihydro-6H-indeno[5,6-e][1,2,4]triazine 1-oxide (***18b***).* Similarly, the reaction of 3-chloride **17b** (0.84 g, 3.79 mmol), glycine methyl ester hydrochloride (0.95 g, 7.59 mmol) and iPr_2_NEt (2.5 mL, 15.19 mmol) in DME (100 mL) gave ester **18b** (0.38 g, 36%) as a yellow powder: mp (MeOH) 182–184 °C; ^1^H NMR δ 8.10 (s, 1 H, H-9), 7.43 (s, 1 H, H-5), 5.65 (t, *J* = 5.1 Hz, 1 H, NH), 4.29 (d, *J* = 5.6 Hz, 2 H, CH_2_CO), 3.80 (s, 3 H, OCH_3_), 2.94–3.05 (m, 4 H, H-6, H-8), 2.15 (t, *J* = 7.4 Hz, 2 H, H-7), 2.15 (pent, *J* = 7.4 Hz, 2 H, H-7); MS *m/z* 275.2 (MH^+^, 100%). Anal. calcd for C_13_H_14_N_4_O_3_·0.2 CH_3_OH: C, 56.48; H, 5.31; N, 19.96. Found: C, 56.24; H, 5.03; N, 20.20%.

*3-((3-Methoxy-3-oxopropyl)amino)-7,8-dihydro-6H-indeno[5,6-e][1,2,4]triazine 1-Oxide (***18c***).* Similarly the reaction of 3-chloro-7,8-dihydro-6*H*-indeno[5,6-*e*][1,2,4]triazine 1-oxide (**17b**) [79] (1.02 g, 4.60 mmol), β-alanine methyl ester (1.28 g, 9.20 mmol) and iPr_2_NEt (3.80 mL, 23.0 mmol) in DME (100 mL) gave the ester **18c** (1.46 g, 79%) as a yellow solid: mp (EtOAc/DCM) 160–161 °C; ^1^H NMR δ 8.08 (s, 1 H, H-9), 7.41 (s, 1 H, H-5), 5.56 (t, *J* = 6.1 Hz, 1 H, NH), 3.81 (dt, *J* = 6.3, 6.1 Hz, 2 H, CH_2_N), 3.71 (s, 3 H, OCH_3_), 2.98–3.06 (m, 4 H, H-6, H-8), 2.71 (t, *J* = 6.1 Hz, 2 H, CH_2_O), 2.15 (pent, *J* = 7.4 Hz, 2 H, H-7); MS *m/z* 298.2 (MH^+^, 100%). Anal. calcd for C_14_H_16_N_4_O_3_: C, 58.32; H, 5.59; N, 19.43. Found: C, 58.41; H, 5.51; N, 19.39%.

*3-((4-Ethoxy-4-oxobutyl)amino)benzo[e][1,2,4]triazine 1-Oxide (***18d***).* Similarly, the reaction of chloride **17a** (1.26 g, 6.94 mmol), ethyl 4-aminobutanoate hydrochloride (1.51 g, 9.00 mmol) and Et_3_N (2.90 mL, 20.8 mmol) in DME (100 mL) gave ester **18d** (1.83 g, 95%) as a yellow powder: mp (EtOAc/pet. ether) 130–131 °C; ^1^H NMR δ 8.26 (dd, *J* = 8.7, 1.4 Hz, 1 H, H-8), 7.70 (ddd, *J* = 8.7, 6.9, 1.4 Hz, 1 H, H-7), 7.60 (br d, *J* = 8.4 Hz, 1 H, H-5), 7.29 (ddd, *J* = 8.4, 6.9, 1.4 Hz, 1 H, H-6), 5.42 (br s, 1 H, NH), 4.15 (q, *J* = 7.1 Hz, 2 H, CH_2_O), 3.58 (dd, *J* = 6.8, 6.2 Hz, 2 H, CH_2_N), 2.44 (t, *J* = 7.2 Hz, 2 H, CH_2_), 2.02 (pent, *J* = 7.0 Hz, 2 H, CH_2_), 1.26 (t, *J* = 7.1 Hz, 3 H, CH_3_); MS *m/z* 277.1 (MH^+^, 100%). Anal. calcd for C_13_H_16_N_4_O_3_: C, 56.51, H, 5.84; N, 20.28. Found: C, 56.78; H, 5.89; N, 20.47%.

*3-((4-Ethoxy-4-oxobutyl)amino)-7-methylbenzo[e][1,2,4]triazine 1-Oxide (***18e***).* Similarly, the reaction of 3-chloro-7-methylbenzo[*e*][1,2,4]triazine 1-oxide (**17b**) [78] (2.65 g, 13.5 mmol), ethyl 4-aminobutanoate hydrochloride (3.41 g, 20.3 mmol) and Et_3_N (5.64 mL, 40.5 mmol) gave ester **18e** (3.68 g, 94%) as a yellow powder: mp (EtOAc) 158–159 °C; ^1^H NMR δ 8.06 (br s, 1 H, H-8), 7.54 (dd, *J* = 8.7, 1.8 Hz, 1 H, H-6), 7.50 (d, *J* = 8.7 Hz, 1 H, H-5), 5.34 (br s, 1 H, NH), 4.15 (q, *J* = 7.1 Hz, 2 H, CH_2_O), 3.57 (dd, *J* = 6.8, 6.2 Hz, 2 H, CH_2_N), 2.46 (s, 3 H, 7-CH_3_), 2.44 (t, *J* = 7.2 Hz, 2 H, CH_2_), 2.02 (pent, *J* = 7.0 Hz, 2 H, CH_2_), 1.24 (t, *J* = 7.1 Hz, 3 H, CH_3_); MS *m/z* 290.1 (MH^+^, 100%). Anal. calcd for C_14_H_18_N_4_O_3_: C, 57.92, H, 6.25; N, 19.30. Found: C, 57.75; H, 6.34; N, 19.38%.

#### 4.2.7. Oxidation of BTO-3-aminoalkanoic Esters (**18a**–**18e**)

*Ethyl [(1,4-Dioxido-1,2,4-benzotriazin-3-yl)amino]acetate (***8b***).* H_2_O_2_ (30%, 10.9 mL, 106 mmol) was added dropwise to a stirred solution of TFAA (16.2 mL, 117 mmol) in DCM (50 mL) at 0–5 °C and the mixture was stirred at 20 °C for 1 h. The mixture was added dropwise to a stirred solution of ester **18a** (2.63 g, 10.6 mmol) in DCM (50 mL) at 0 °C and the mixture was stirred at 20 °C for 5 days. The mixture was cooled to 0 °C and DMSO (7.5 mL, 106 mmol) was added. The mixture was carefully neutralized with a concentrated NH_3_ solution (ca. 15 mL) and the mixture was washed with water (3 × 30 mL), dried and the solvent was evaporated. The residue was purified by chromatography, eluting with a gradient (0–10%) of MeOH/EtOAc, to give (i) starting material **18a** (593 mg, 23%), spectroscopically identical with the sample above; and (ii) di-oxide **8b** (1.535 g, 55%) as a red powder: mp (EtOAc/pet. ether) 156–158 °C; ^1^H NMR δ 8.34 (d, *J* = 8.7 Hz, 2 H, H 5, H 8), 7.91 (br s, 1 H, NH), 7.50–7.61 (m, 2 H, H 6, H 7), 4.37 (br s, 2 H, CH_2_), 4.26 (q, *J* = 7.1 Hz, 2 H, CH_2_), 1.30 (t, *J* = 7.1 Hz, 3 H, CH_3_); ^13^C NMR δ 168.5, 149.7, 138.4, 136.0, 131.0, 129.6, 121.7, 117.6, 61.9, 42.8, 14.1; MS (EI^+^) *m/z* 264 (M^+^, 30%), 284 (40), 175 (100); HRMS (EI^+^) calc. for C_11_H_12_N_4_O_4_ (M^+^) *m/z* 264.0859, found 264.0852. Anal. calcd for C_11_H_12_N_4_O_4_: C, 50.0; H, 4.6; N, 21.2. Found: C, 50.8; H, 4.5; N, 21.1%.

*3-((2-Methoxy-2-oxoethyl)amino)-7,8-dihydro-6H-indeno[5,6-e][1,2,4]triazine 1,4-dioxide (***9b***).* Similarly, the reaction of H_2_O_2_ (30%, 1.4 mL, 14.1 mmol) and TFAA (1.8 mL, 12.8 mmol) with ester **18b** (350 mg, 1.28 mmol) gave di-oxide **9b** (86 mg, 23%) as a red powder: mp (MeOH/EtOAc) 171–178 °C; ^1^H NMR δ 8.14 (s, 1 H, H-9), 8.13 (s, 1 H, H-5), 7.40 (br s, 1 H, NH), 4.36 (d, *J* = 5.9 Hz, 2 H, CH_2_N), 3.80 (s, 3 H, OCH_3_), 3.13 (t, *J* = 7.2 Hz, 2 H, H-8), 3.07 (dt, *J* = 7.4, 1.0 Hz, 2 H, H-6), 2.12 (p, *J* = 7.5 Hz, 2 H, H-7); ^13^C NMR δ 169.3, 156.1, 149.3, 146.9, 138.2, 130.6, 116.0, 112.0, 52.9, 43.0, 33.7, 32.6, 25.8; MS *m/z* 291.2 (MH^+^, 100%). Anal. calcd for C_13_H_14_N_4_O_4_: C, 53.79; H, 4.86; N, 19.30. Found: C, 53.84; H, 4.79; N, 19.41%. HPLC purity 96.4%.

*3-((3-Methoxy-3-oxopropyl)amino)-7,8-dihydro-6H-indeno[5,6-e][1,2,4]triazine 1,4-Dioxide (***10b***).* Similarly, the reaction of H_2_O_2_ (30%, 1.25 mL, 10.0 mmol) and TFAA (1.26 mL, 9.1 mmol) with ester **18c** (249 mg, 0.91 mmol) gave di-oxide **10b** (200 mg, 76%) as a yellow powder: mp (EtOAc/pet. ether) 170–171 °C; ^1^H NMR δ 8.31 (s, 1 H, H-9), 8.25 (s, 1 H, H-5), 3.71 (s, 3 H, OCH_3_), 3.48 (t, *J* = 7.3 Hz, 2 H, CH_2_), 3.12–3.20 (m, 4 H, H-6, H-8), 2.97 (t, *J* = 7.2 Hz, 2 H, CH_2_CO), 2.26 (p, *J* = 7.5 Hz, 2 H, H-7); ^13^C NMR δ 172.6, 155.4, 153.2, 151.0, 139.2, 134.1, 116.1, 114.0, 52.2, 33.6, 33.1, 28.9, 25.8, 25.7; MS *m/z* 290.3 (MH^+^, 100%). Anal. calcd for C_14_H_15_N_3_O_4_: C, 58.13; H, 5.23; N, 14.53. Found: C, 57.91; H, 5.20; N, 14.44%. HPLC purity 99.3%.

*3-((4-Ethoxy-4-oxobutyl)amino)benzo[e][1,2,4]triazine 1,4-Dioxide (***11b***).* Similarly, the reaction of H_2_O_2_ (30%, 6.6 mL, 64.4 mmol) and TFAA (9.6 mL, 70.9 mmol) in DCM (40 mL) with ester **18d** (1.78 g, 6.44 mmol) gave starting ester **18d** (751 mg, 42%), spectroscopically identical to above, and di-oxide **11b** (355 mg, 19%) as a red powder: mp (EtOAc) 163–165 °C; ^1^H NMR δ 8.34 (dd, *J* = 8.7, 0.8 Hz, 1 H, H-8), 8.30 (dd, *J* = 8.7, 1.3 Hz, 1 H, H-5), 7.88 (ddd, *J* = 8.7, 7.0, 1.3 Hz, 1 H, H-7), 7.52 (ddd, *J* = 8.7, 7.0, 1.0 Hz, 1 H, H-6), 7.30 (br s, 1 H, NH), 4.16 (q, *J* = 7.1 Hz, 2 H, CH_2_O), 3.67 (dd, *J* = 6.8, 6.6 Hz, 2 H, CH_2_N), 2.46 (t, *J* = 7.2 Hz, 2 H, CH_2_), 2.08 (pent, *J* = 7.1 Hz, 2 H, CH_2_), 1.26 (t, *J* = 7.1 Hz, 3 H, CH_3_); ^13^C NMR [(CD_3_)_2_SO] δ 172.9, 150.0, 138.4, 136.0, 130.7, 127.4, 121.8, 117.5, 60.8, 41.0, 31.5, 24.7, 14.3; MS *m/z* 293.2 (MH^+^, 100%). Anal. calcd for C_13_H_16_N_4_O_4_: C, 53.42; H, 5.52; N, 19.17. Found: C, 53.41; H, 5.49; N, 19.40%. HPLC purity 99.9%.

*3-((4-Ethoxy-4-oxobutyl)amino)-7-methylbenzo[e][1,2,4]triazine 1,4-Dioxide (***12b***).* Similarly, the reaction of H_2_O_2_ (30%, 5.0 mL, 48.2 mmol) and TFAA (7.4 mL, 53.0 mmol) with ester **18e** (1.40 g, 4.82 mmol) gave starting ester **18b** (586 mg, 42%), spectroscopically identical to above, and di-oxide **12b** (202 mg, 14%) as a red powder: mp (EtOAc) 191–192 °C; ^1^H NMR δ 8.19 (d, *J* = 8.9 Hz, 1 H, H-5), 8.13 (br s, 1 H, H-8), 7.70 (dd, *J* = 8.9, 1.6 Hz, 1 H, H-6), 7.13 (br s, 1 H, NH), 4.16 (q, *J* = 7.1 Hz, 2 H, CH_2_O), 3.65 (dd, *J* = 7.0, 5.7 Hz, 2 H, CH_2_N), 2.53 (s, 3 H, 7-CH_3_), 2.45 (t, *J* = 7.2 Hz, 2 H, CH_2_), 2.06 (pent, *J* = 7.0 Hz, 2 H, CH_2_), 1.26 (t, *J* = 7.1 Hz, 3 H, CH_3_); ^13^C NMR [(CD_3_)_2_SO] δ 173.0, 149.7, 138.6, 138.3, 137.0, 130.6, 120.4, 117.3, 60.9, 41.0, 31.6, 24.7, 21.6, 14.4; MS *m/z* 307.2 (MH^+^, 100%). Anal. calcd for C_14_H_18_N_4_O_4_: C, 54.89; H, 5.92; N, 18.29. Found: C, 55.03; H, 5.77; N, 18.50%. HPLC purity 99.9%.

#### 4.2.8. Hydrolysis of Esters (**8b**–**12b**)

*3-((Carboxymethyl)amino)benzo[e][1,2,4]triazine 1,4-dioxide (***8a***).* A solution of HCl in dioxane (4 M, 69 mL, 277 mmol) was added dropwise to a stirred suspension of ester **8b** (1.44 g, 5.55 mmol) in dioxane (40 mL) and water (5 drops), and the mixture was stirred at 20 °C for 6 d. The mixture was diluted with water (50 mL) and extracted with CHCl_3_ (3 × 60 mL), then the combined organic fraction was dried and the solvent was evaporated. The residue was purified by chromatography, eluting with a gradient (0–5%) of MeOH/DCM, to give (i) starting ester **8b** (570 mg, 40%), spectroscopically identical to above; and (ii) acid **8a** (99 mg, 8%) as a red powder, mp (H_2_O) 190–191 °C; ^1^H NMR [(CD_3_)_2_SO] δ 8.45 (br s, 1 H, NH), 8.23 (d, *J* = 8.7 Hz, 1 H, H 8), 8.19 (d, *J* = 8.5 Hz, 1 H, H 5), 7.99 (ddd, *J* = 8.5, 7.0, 1.3 Hz, 1 H, H 6), 7.63 (ddd, *J* = 8.7, 7.0, 1.3 Hz, 1 H, H 7), 4.09 (d, *J* = 5.4 Hz, 2 H, CH_2_), OH not observed; ^13^C NMR [(CD_3_)_2_SO] δ 170.5, 149.8, 138.2, 135.8, 130.3, 127.6, 121.1, 117.0, 42.3. Anal calcd for C_9_H_8_N_4_O_4·_¼H_2_O; C, 44.9; H, 3.6; N, 23.3. Found: C, 44.5; H, 3.4; H, 23.4%. HPLC purity 98.4%.

*3-((3-Carboxypropyl)amino)benzo[e][1,2,4]triazine 1,4-Dioxide (***10a***).* Similarly, the reaction of HCl in dioxane (4 M, 13 mL, 52 mmol) with ester **10b** (298 mg, 1.02 mmol) in dioxane (10 mL) and water (0.5 mL) gave starting ester **10b** (36 mg, 12%), spectroscopically identical to above, and acid **10a** (158 mg, 59%) as an orange powder: mp (EtOAc) 190–193 °C; ^1^H NMR [(CD_3_)_2_SO] δ 12.06 (br s, 1 H, CO_2_H), 8.35 (t, *J* = 6.2 Hz, 1 H, NH), 8.19 (dd, *J* = 8.7, 0.7 Hz, 1 H, H-8), 8.12 (dd, *J* = 8.7, 0.7 Hz, 1 H, H-5), 7.92 (ddd, *J* = 8.7, 6.9, 1.2 Hz, 1 H, H-7), 7.55 (ddd, *J* = 8.7, 6.9, 1.2 Hz, 1 H, H-6), 3.43 (q, *J* = 6.4 Hz, 2 H, CH_2_N), 2.30 (t, *J* = 7.4 Hz, 2 H, CH_2_), 1.84 (pent, *J* = 7.1 Hz, 2 H, CH_2_); ^13^C NMR [(CD_3_)_2_SO] δ 174.2, 149.8, 138.2, 135.4, 129.9, 126.9, 121.1, 116.9, 40.1, 31.0, 24.1; MS *m/z* 265.2 (MH^+^, 100%). Anal. calcd for C_11_H_12_N_4_O_4_: C, 50.00; H, 4.58; N, 21.20. Found: C, 50.31; H, 4.54; N, 20.82%. HPLC purity 98.6%.

*3-((2-Carboxyethyl)amino)-7,8-dihydro-6H-indeno[5,6-e][1,2,4]triazine 1,4-Dioxide (***11a***).* Similarly the reaction of HCl in dioxane (4 M, 4.32 mL, 17.3 mmol) with ester **11b** (100 mg, 0.34 mmol) gave starting ester **11b** (13 mg, 13%), spectroscopically identical to above, and acid **11a** (60 mg, 64%) as a yellow powder: mp (EtOAc) 159 °C (dec.); ^1^H NMR δ 12.34 (br s, 1 H, CO_2_H), 8.18 (s, 1 H, H-9), 8.17 (s, 1 H, H-5), 3.22 (t, *J* = 7.2 Hz, 2 H, CH_2_), 3.07–3.16 (m, 4 H, H-6, H-8), 2.76 (t, *J* = 7.2 Hz, 2 H, CH_2_CO), 2.13 (p, *J* = 7.5 Hz, 2 H, H-7); ^13^C NMR δ 173.2, 154.6, 152.7, 150.2, 138.7, 133.5, 115.2, 113.1, 32.8, 32.3, 28.5, 25.2 (2); MS *m/z* 276.3 (MH^+^, 100%). Anal. calcd for C_13_H_13_N_3_O_4_: C, 56.72; H, 4.76; N, 15.27. Found: C, 57.00; H, 4.73; N, 15.08%. HPLC purity 99.7%.

*3-((3-Carboxypropyl)amino)-7-methylbenzo[e][1,2,4]triazine 1,4-Dioxide (***12a***).* Similarly, the reaction of HCl in dioxane (4 M, 3.5 mL, 13.8 mmol) with ester **12b** (85 mg, 0.27 mmol) gave starting ester **12b** (36 mg, 12%), spectroscopically identical to above, and acid **12a** (50 mg, 66%) as a red powder: mp (MeOH/H_2_O) 185–188 °C; ^1^H NMR [(CD_3_)_2_SO] δ 12.06 (br s, 1 H, CO_2_H), 8.26 (t, *J* = 6.1 Hz, 1 H, NH), 8.04 (d, *J* = 8.9 Hz, 1 H, H-5), 8.01 (br s, 1 H, H-8), 7.77 (dd, *J* = 8.9, 1.5 Hz, 1 H, H-6), 3.40 (q, *J* = 6.7 Hz, 2 H, CH_2_N), 2.47 (s, 3 H, 7-CH_3_), 2.30 (t, *J* = 7.4 Hz, 2 H, CH_2_), 1.85 (pent, *J* = 7.1 Hz, 2 H, CH_2_); ^13^C NMR [(CD_3_)_2_SO] δ 174.3, 149.5, 137.5, 137.4, 136.8, 129.7, 119.5, 116.8, 40.2, 31.1, 24.1, 20.8; MS *m/z* 278.2 (MH^+^, 100%). Anal. calcd for C_12_H_14_N_4_O_4_·¼H_2_O: C, 50.97; H, 5.17; N, 19.81. Found: C, 50.73; H, 5.04; N, 19.59%. HPLC purity 98.5%.

### 4.3. Measurement of pKa of **7a**

The pKa of **7a** was determined by Sirius Analytical (Forest Row, UK) by potentiometric triple titration from pH 12.0 to 2.0, using three solutions in 0.15 M of KCl at concentrations of 1.2–0.9 mM. No precipitation of the sample from solution was observed and one pKa with an average value of 4.16 ± 0.01 was determined.

### 4.4. Measurement of One-Electron Reduction Potentials, E^0′^

Time-resolved optical absorption and kinetic studies were carried out using the University of Auckland’s 4 MeV linear accelerator, which has been previously described [80]. The one-electron reduction potentials of the compounds (A) and couple *E*^0′^(A/A**^−^**), versus normal hydrogen electrode, were determined at pH 7.0 (2.5 mM phosphate) by establishing redox equilibria within 50 μs between three mixtures of one-electron reduced A and the viologen reference compounds, MV^2+^, TQ^2+^ and V21^2+^. The determined equilibrium constants were used to calculate the Δ*E* values using the Nernst equation, correcting for ionic strength effects, as described in the literature [81].

### 4.5. EPR Characterization of the **7a**-Derived Radical Intermediates

The radical intermediates formed following the reduction of **7a** by N-terminally truncated (soluble) POR [24] under anoxia were spin-trapped and identified using a JOEL (JES-FA-200) EPR spectrophotometer. The solution preparation procedure and spectrophotometer settings have been previously described [82]. Computer simulation of the obtained composite spectrum was carried out using the WINSIM EPR program, available in the public domain of the NIEHS EPR database.

### 4.6. Calculation of pH Dependence of logP and Cellular Uptake (Ci/Ce Ratios)

Octan-1-ol/water partition coefficients for the neutral and charged forms of the BTO acids (*P_n_* and *P_c_* respectively) were calculated in ChemDraw (v17.1.0 PerkinElmer Informatics Inc., Cambridge, MA, USA). The mole fraction of the neutral and charged species (*f_n_* and *f_c_*) were determined using the Henderson-Hasselbalch equation. LogD values were calculated at each pH using:(1)$$logD=log\left(f_{n}P_{n}+f_{c}P_{c}\right)$$

Ci/Ce values for pH-dependent partitioning were calculated using Equation (2) for weak acids and Equation (3) for the weak base SN30000 [12]:(2)$$\frac{C_{i}}{C_{e}}=\left(\frac{10^{-pH_{i}}+10^{-pK_{a}}}{10^{-pH_{e}}+10^{-pK_{a}}}\right)\times 10^{\left(pH_{i}-pH_{e}\right)}$$

(3)$$\frac{C_{i}}{C_{e}}=\left(\frac{10^{-pH_{i}}+10^{-pK_{a}}}{10^{-pH_{e}}+10^{-pK_{a}}}\right)$$

### 4.7. Cell Lines

The SiHa and FaDu cell lines were from ATCC and were authenticated in-house by short tandem repeat profiling. Cultures were maintained in αMEM (ThermoFisher) with 5% FBS (Moregate Biotech, Hamilton, NZ) and passaged for up to 3 months without antibiotics. The SiHa/POR line, reported previously [83], was maintained in the same medium with 1 μM puromycin. UT-SCC-54C was sourced, characterized, and maintained in culture as reported [84]. All lines were confirmed to be mycoplasma free using PlasmoTest™ (InvivoGen).

### 4.8. Titration of Media and Measurement of Extracellular pH (pHe)

The pH of the cell culture medium was determined using an Orion A211 pH meter with a Thermo 3 mm diameter pH microprobe (Fisher Scientific) under an atmosphere of 5% CO_2_ at 37 °C. The pH of αMEM containing 10% FBS was titrated to pH 6.5 with concentrated HCl while stirring magnetically at 37 °C under flowing humidified 5% CO_2_. The HCl was added in small aliquots over approximately 6 h to allow HCO_3_^−^/CO_2_ re-equilibration whilst maintaining a pH above 6.0 throughout.

### 4.9. Determination of Intracellular pH (pHi)

For the measurement of pHi under oxia (20% O_2_), cells were seeded at 2 × 10^5^/well in 0.2 mL αMEM + 10% FBS per well in 96-well plates. After incubation at 37 °C for 18 h in a humidified 5% CO_2_/20% O_2_ incubator, the monolayers were washed with a fresh medium to restore the pHe to 7.4, after which they were incubated for a further 60 min and washed with serum-free Hank’s balanced salt solution (HBSS; ThermoFisher 14065) containing 5.6 mM glucose that had been titrated to pH 7.4 or 6.5, as in Section 4.8. The cells were washed with HBSS at pH 7.4 or 6.5, then loaded with BCECF AM (2 μM; Molecular Probes, Invitrogen B1150) in the same buffers for 30 min. To ensure the control of temperature and pHe during this loading step, the plates were incubated on an Eppendorf Thermomixer, with flowing humidified 5% CO_2_/air under a loose-fitting lid. After the 30 min loading period, fluorescence (excitation of 440 and 490 nm, emission of 535 nm) in this plate (plate A) was immediately measured with an Enspire multimode plate reader (Perkin Elmer). The extracellular medium was then transferred to a second plate (plate B), where fresh HBSS at the same pH was added to plate A and the fluorescence in both plates was determined with the Enspire platereader. The difference in the pH-independent fluorescence intensity at 440 nm between plate A (post-wash) and plate B indicated that typically ~80% of the BCECF in plate A was intracellular pre-wash, but only the post-wash measurements for plate A were used in order to minimize any contribution of extracellular BCECF (incubation for >40 min post-loading resulted in the progressive loss of intracellular BCECF, especially at pHe 6.5, so only the initial timepoint was used for calculation of pHi).

Calibration was achieved by loading with BCECF AM in HBSS at pHe 7.4 as above, in separate plates, washed twice with HBSS then solutions of digitonin (Serva) in PBS, where titrations to pH 6.0, 6.5, 7.0, 7.5, or 8.0 were added at 0.2 mL/well. The fluorescence was measured as above, then the pH in each well was measured with the pH microprobe and the fluorescence ratio (490/440) versus the pH fitted by linear regression.

For the measurement of pHi under hypoxia (0.2% O_2_), plates were transferred to a hypoxystation (Model H45, Don Whitley Scientific) with a humidified gas phase of 0.2% O_2_/5% CO_2_/balance N_2_ at 37 °C. Twenty-four hours later, cells were loaded with BCEFC AM for 30 min and the pHi was determined as above, with the measurement of fluorescence immediately after removing the plates from the hypoxystation.

For the measurement of pHi under anoxia (<0.01% O_2_), cells were pelleted by centrifugation, transferred to a Pd/C anaerobic chamber (5% CO_2_, 5% H_2_, 90% H_2_; Shellab, Sheldon Manufacturing Inc. Cornelius, OR), resuspended in αMEM + 10% FBS, and plated as above, using medium and 96-well plates equilibrated in the chamber for 3 days prior to use to remove residual O_2_. The cells were loaded with BCECF AM as above, with the probe added 4 h after plating (equivalent to the midpoint of the drug exposures in the IC_50_ assays of Section 4.10).

### 4.10. IC_50_ Assays

Cells in log-phase growth were trypsinized, collected by centrifugation, and then resuspended in AlphaMEM with 10% FBS, additional glucose (final concentration 15.6 mM), 2′deoxycytidine (20 μM), penicillin, and streptomycin (“IC50 medium”) that had been titrated to pH 7.4 or 6.5, as above. Cells were seeded in 96-well plates (SiHa or FaDu 1500 cells/well, UT-SCC-74B 800 cells/well) in 100 μL. After incubation for 2 h, the drugs were added in 50 μL of the same titrated medium at 3× the final concentration to duplicate the wells. After a further 4 h the plates were checked for drug precipitation using a phase contrast microscope, then drugs were removed by aspirating and washing with the standard (pH 7.4) medium, then aspirating again and adding 150 μL of the medium. Cultures were then grown for 5 days before staining with sulphorhodamine B and measuring absorbance, with the estimation of IC_50_ values using 4-parameter logistic regression, as previously carried out [85].

For drug exposures under anoxia (<0.01% O_2_ gas phase), the pellets were transferred to a Pd/C anaerobic chamber as above, resuspended in IC50 medium, and exposed as above with all steps until the end of drug treatment performed in the chamber, following which the cells were regrown under 20% O_2_. The medium and plasticware was equilibrated in the chamber for 3 days prior to use. In the anoxia experiments, FaDu cells were plated in poly-L-lysine coated plates (Corning Cat # 356461) to improve attachment.

For drug exposures under chronic hypoxia (0.2% O_2_ in the Whitley hypoxystation), the following modifications to the above methods were made: Cells were seeded in the hypoxia chamber at 1125 (SiHa) or 600 (UT-SCC-74B) cells/well in the IC50 medium 24 h before drug exposure. To minimize problems with poor cell attachment under chronic hypoxia, the drugs were added from a 3-fold dilution series made in a separate plate, rather than by serial dilution in the plates containing the cells. In addition, the drugs were removed after the 4 h exposure by centrifuging the plates (200*g* × 5 min) prior to each aspiration. The same modifications were used for the oxic drug exposures in these experiments.

### 4.11. Drug Metabolism and Cellular Uptake

#### 4.11.1. Cellular Uptake and Metabolism in Single Cell Suspensions

Log-phase cultures were dissociated with trypsin/EDTA, followed by the addition of the medium with DNAase I (Sigma DN-25) at 100 μg/mL for 5 min at 37 °C to minimize clumping. The cell pellets were collected by centrifugation and resuspended in αMEM with 5% FBS at pH 7.4 or 6.5 to 2 × 10^6^ cells/mL. Samples (12 mL) were equilibrated with magnetic stirring at 37 °C under flowing humidified 5% CO_2_/air or N_2_ (50–100 mL/min), as previously described [86]. After equilibration for 60 min, the compounds were added in 50 μL of DMSO. The samples (5 mL) were withdrawn 5 and 60 min later, where the cell pellets were then collected by centrifugation and 4 mL of the supernatant was discarded. The pellets were resuspended in the remaining 1 mL, transferred to microcentrifuge tubes containing 7 μL ^3^H-mannitol (740 GBq/mmol, American Radiolabeled Chemicals Inc., St. Louis, MO) as an extracellular marker, and then centrifuged (11,000*g* × 30s). The supernatant was removed and frozen at −80 °C (extracellular samples) and the tubes were centrifuged again, and any remaining medium was carefully removed. Ice-cold MeOH (80 μL) was added to the pellets which were disrupted by pipetting and stored at −80 °C (intracellular samples). When these were thawed for HPLC analysis (Section 4.12), the volume of extracellular water trapped in the pellets was determined by the scintillation counting (PerkinElmer Tri-Carb B2910TR Liquid Scintillation) of 25 μL of each extracellular and intracellular sample in 3 mL of Emulsifier-Safe™ scintillation cocktail (PerkinElmer). The extracellular volume in each intracellular sample was used to correct the intracellular drug concentration as previously described [87].

#### 4.11.2. Cellular Metabolism by Anoxic Monolayers in 96-Well Plates

The log-phase cells were trypsinized and the pellets were transferred to the anaerobic chamber and resuspended in equilibrated αMEM + 5% FBS at pH 7.4 and plated (10^5^ cells in 100 μL/well for experiments with WST1, otherwise 2 × 10^5^ cells/well). After incubation at 37 °C for 2 h, 20 μL was replaced with the anoxic medium containing 5× the final concentrations of test compounds with or without WST-1 (1 mM, Toronto Research Chemicals). A cell-free medium used as a control to check for reduction in the anoxic medium. Cultures were mixed gently, incubated for a further 2 h, then the extracellular medium was removed and 200 μL of ice-cold MeOH was added to the monolayers. The MeOH extract was transferred to the medium sample (thus combining the intracellular and extracellular samples). The pooled samples (combining the intracellular and extracellular samples) were removed from anoxic chamber, then centrifuged (13000*g*, 30 s) and frozen for subsequent HPLC analysis. Cell-free wells were used for single-point calibration of the parent compounds and 1-oxides in each experiment.

#### 4.11.3. Metabolism by Post-Mitochondrial Supernatants (S9 Preparations)

The S9 fractions were prepared from approximately 5 × 10^7^ log-phase SiHa and SiHa/POR cells, as described previously [88]. The protein concentrations were determined with the bicinchoninic acid assay. The S9 preparations were diluted to 50 μg/mL in 1 mM of NADH, 1 mM of NADPH, 7.1 mM of MgCl_2_, 1.4 mM of Na_2_EDTA, and a 67 mM phosphate buffer (pH 7.0) in 96-well plates within the anaerobic chamber. After equilibration at 37 °C for 30 min, 10 μL DMSO solutions of the compounds were added. After incubation for a further 30 min the plates were removed from the chamber and the proteins were precipitated with 200 μL of ice-cold MeOH and then frozen for HPLC analysis.

### 4.12. HPLC Analysis

The samples were analyzed using an Agilent 1200 HPLC with a diode array absorbance detector and autosampler maintained at 4 °C (Agilent Technologies). The frozen samples were thawed, then centrifuged (11,000*g* × 5 min), and then extracellular samples were injected directly (typically 20 μL), while for the intracellular samples 30 μL was mixed 1:1 with αMEM and 50 μL was injected. The calibration standards for the parent compounds and 1-oxides were prepared freshly in the culture medium (typically 0.3-100 μM) for each analysis. All calibration curves were linear (R^2^ > 0.999). The column was an Agilent ZORBAX Eclipse (XDB-C18, 5u, 2.1 × 150 mm, Agilent Technologies) with a flow rate of 0.4 mL/min. For SN30000 (**2**), the mobile phase was a mixture of 80% MeCN/water *v*/*v* (A), and a 45 mM formate buffer at a pH 4.5, eluting with a linear gradient of 5–100% A over 18 min. The same method was used for the BTO acids, except for **4a**, where the aqueous mobile phase pH was reduced to 3.5. The analytes were quantified using the following wavelengths: 294 nm for SN30000, with a retention time (Rt) of 8.6 min; 404 nm for SN30000 1-oxide, with a Rt of 9.6 min; 400 nm for **4a**, with a Rt of 6.5 min; 360 nm for **15a**, with a Rt of 8.8 min; 296 nm for **7a**, with a Rt of 10.5 min; and 250 nm for **15d**, with a Rt 13.1 min.

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
