# Peer review of "Benzotriazine Di-Oxide Prodrugs for Exploiting Hypoxia and Low Extracellular pH in Tumors"

_molecules, 2019, doi:10.3390/molecules24142524_

Round 1
Reviewer 1 Report
This is a very interesting study and the work has been thoroughly conducted, analysed and presented. I only have a few very minor comments to make (i) it would be useful if the TME ratio were illustrated on figure 3c - this would marry up nicely with the PCR and ACR parameters as illustrated in figure 3A and 3C (ii) page 5, line 145: .....ruling out the hydroxyl radical is being ..... I would remove the word 'is' from this sentence (iii) table 2, why were some studies n=1 or n = 2?
Author Response
(i) TME ratio is now illustrated, using the abbreviation TMR, on Figure 3c. This abbreviation is now used consistently throughout.
(ii) The sentence on line 145 has been corrected.
(iii) The main objectives of the pHi investigation were to establish the magnitude of the pHi shift when cultures are acidified, and whether pHi regulation is altered under chronic hypoxia which could occur through know HIF-1 upregulation of pHi regulating transporters. There is no reason to expect acute anoxia to have such effects so we didn’t investigate beyond a single experiment which confirmed this. However, there were 3 replicate biological replicates within this experiment (which we have now indicated in the footnotes to Table 2) so the result is reasonably robust.
Reviewer 2 Report
This manuscript entitled “Benzotriazine di-oxide prodrugs for exploiting hypoxia and low extracellular pH in tumors" deals with the concept of dual targeting to cancer with hypoxia and low pH environment. Unfortunately, BTO acids in this study have poor anticancer effects by several unapparent reasons, however, this concept may be expected in the future.
The authors synthesized and tested eight BTO acids, then I received the impression that the compound of interest was different depending on each experiment. For example, Figure 2 focuses on 7a, Figure 3, 7 and Table 7 focuses on 4a, Figure 4 and Table 8 focuses on 4a, 7a, and 11a, and Figure 5, 6, 8 focuses on 4a, 7a. On page 7, line 212 discussed 4a, 7a, 10a but not 11a. I think that the explanation and discussion are insufficient to explain why the authors focused on their compounds.
On page 7, line 212: "ranged from 140 to 3800 uM"; Is IC50 value of 4a (97uM) under anoxia at pHe6.5 excluded?
On page 10, line 269: "in the range 46-156"; Is this value (46) correct?
Table 8: It's my guess that all the Obs of 4a in Table 8 cannot be calculated because of intracellular 4a was ND in Figure 4.
On page 7, line 202: Please define abbreviation (CHL).
Figure S9, line 1: Please fix typo (phenypyruvate to phenylpyruvate).
On page 22, line 650: Please fix typo (EtOAC to EtOAc).
On page 24, line 766 and 779: Please make 17a and 17b bold.
Author Response
Author responses are in red.
I think that the explanation and discussion are insufficient to explain why the authors focused on their compounds.
This paper is already extremely data rich and it would become very cumbersome if all assays were performed on all compounds. For the growth inhibition data, which are central to the paper, we illustrate the assays with representative compounds (Figure 2, using compounds 1, 2 and 4a), but have presented all the data for the compounds in these assays in Tables 3-6. For the EPR experiments, which are considerable undertakings, a single compound with a one-electron reduction potential that allows ready enzymatic reduction was selected as a representative example. For the experiments reported in the other figures, our approach was to pick several examples of compounds which do display significant PCR and ACR values, and for which solubility/potency was not limiting, to explore and document the mechanisms that are contributing to the observed activity.
On page 7, line 212: "ranged from 140 to 3800 uM"; Is IC50 value of 4a (97uM) under anoxia at pHe6.5 excluded?
No it should have been included. Corrected.
On page 10, line 269: "in the range 46-156"; Is this value (46) correct?
Yes, this range include all the ACR values reported for compound 2 against SiHa cells: 46 from Table 3, and 67 and 156 from Table 7.
Table 8: It's my guess that all the Obs of 4a in Table 8 cannot be calculated because of intracellular 4a was ND in Figure 4.
Yes that is correct.
On page 7, line 202: Please define abbreviation (CHL). Defined at first use on page 1 line 52.
Figure S9, line 1: Please fix typo (phenypyruvate to phenylpyruvate). Corrected
On page 22, line 650: Please fix typo (EtOAC to EtOAc). Corrected
On page 24, line 766 and 779: Please make 17a and 17b bold. Corrected
Reviewer 3 Report
Manuscript molecules-541915 entitled "Benzotriazine di-oxide prodrugs for exploiting hypoxia and low extracellular pH in tumors" by Hay M.P., describes the synthesis, the biological evaluation and some pharmacokinetic investigations of a set of benzotriazine di-oxide derivatives. These molecules show a carboxylic acid moiety which should exploit the pH gradient across the plasma membrane in cancer cells to accumulate into cells. Also the corresponding methyl or ethyl esters were evaluated. Unfortunately, acid BTO derivatives, were poorly active while better results were obtained by the corresponding esters. Based on the reported experiments, authors suppose that these results are due to the extracellular extensive metabolism which inactivate acid BTO derivatives to the inactive corresponding 1-oxides.
In my opinion this is a good work, the rationale behind the reported experiments is quite clear, as well as the analysis of the obtained results. I have only to note that it is not appropriate to underline BTO acids selectivity for pH conditions, as reported in the abstract (lines 20-21) and at the beginning of the Discussion section (lines 471-472), as these compounds are substantially inactive and the selectivity degree is very low (<10 folds). Therefore these statements should be removed or changed.
Author Response
The sentence “The BTO acids showed significant selectivity for both low pHe (pH 6.5 versus 7.4, ratios 2 to 5-fold) and anoxia (ratios 2 to 8-fold) in SiHa and FaDu cell cultures.” is correct and when compared to CHL the ratios (8.06 and 4.7) are not dissimilar. Given that CHL has been touted as an example of a pH-selective compound for decades, as we discuss in the Introduction, the selectivities observed for the BTO acids still deserve noting. Nonetheless, we appreciate the reviewer’s perspective and we have altered the text in the abstract from “significant selectivity” to “modest selectivity”. Similarly we have amended the sentence in the discussion to read “Neutral (BTO esters and TPZ) or weakly basic BTOs (SN30000) lack the modest pH selectivity of the BTO acids, but it is notable that the latter are much less potent under all conditions tested.”
Reviewer 4 Report
In the manuscript entitled “Benzotriazine di-oxide prodrugs for exploiting hypoxia and low extracellular pH in tumors”, William R. Wilson et al. presented their comprehensive studies on di-oxide designed to selectively act as anti-tumor agents under hypoxic and acidic conditions characteristic to tumor tissue. The authors prepared a number of benzotriazine di-oxide derivatives and characterized deeply their physico-chemical properties. Unfortunately, the new prodrugs appeared to be less efficient than three related known anti-tumor compounds. The authors performed excellent comprehensive biochemical studies to understand this behavior and drew a logical hypothesis on metabolism of the studied species.
Only a few minor corrections or additions are required:
1. In Scheme 1, a “no-go” arrow from 15 to 4-7a, with conditions specified would be advantageous.
2. In Schemes 1 and 2 yields should be shown for intermediate and final products. Having all yields in one place will be convenient for the readers.
3. Line 169: “Similar results were obtained with cells under hypoxia (0.2% O2)” – this is clearly not true. While ΔpHi is similar, indeed, pHi’s are significantly different. This should be corrected.
In summary, I recommend publishing the manuscript.
Author Response
Author responses are in red.
1. In Scheme 1, a “no-go” arrow from 15 to 4-7a, with conditions specified would be advantageous. The Scheme has been adapted to include the no-go arrow and the experimental conditions are included in the legend.
2. In Schemes 1 and 2 yields should be shown for intermediate and final products. Having all yields in one place will be convenient for the readers. Yields have been included in the Schemes
3. Line 169: “Similar results were obtained with cells under hypoxia (0.2% O2)” – this is clearly not true. While ΔpHi is similar, indeed, pHi’s are significantly different. This should be corrected. In fact the differences are not statistically significant, which we have now explained in the text (line 169-170).